

# Precipitation alters plastic film mulching impacts on soil

# respiration in an arid area of Northwest China

Guanghui Ming[1], Hongchang Hu[1], Fuqiang Tian[1*], Zhenyang Peng[1], Pengju Yang[1], Yiqi Luo[2, 3],

[1]Department of Hydraulic Engineering, State Key Laboratory of Hydroscience and Engineering, Tsinghua University, Beijing 100084, China,

[2]Department of Earth System Science, Tsinghua University, Beijing 100084, China

[3]Department of Microbiology and Plant Biology, University of Oklahoma, Norman, Oklahoma, USA



**Abstract:** Plastic film mulching (PFM) has been widely used for saving water and improving yield around the world, particularly in arid areas. However, the effect of PFM in agriculture on soil respiration is still unclear, and this effect may be confounded with irrigation and precipitation. To detect the effects of PFM, irrigation and precipitation on the temporal and spatial variations in soil respiration, plastic mulched and non-mulched drip irrigation contrast experiments were conducted in the arid area of the Xinjiang Uygur Autonomous Region, Northwest China. PFM generated more complicated spatial heterogeneity in the microclimate with increased albedo, improved soil temperature, soil moisture and crop growth, and led to the stronger spatial heterogeneity of the soil respiration. The soil respiration in the plant holes was larger than in the furrows, and plastic mulch itself can emit up to 2.75 μmol $m^{-2}$ $s^{-1}$ $CO_2$, which indicates that furrows, plant holes and plastic mulch were the important pathways for $CO_2$ emissions in the mulched field. Frequent irrigation and precipitation made the soil respiration much more dynamic and fluctuated. The sensitivity of the soil respiration to soil temperature was weakened by extreme variations in the soil moisture with lower correlation and $Q_{10}$ values. In the wetting-drying cycle, both irrigation and precipitation restrained the soil respiration at a high soil water content (SWC) with a threshold of 60% water-filled pore space (WFP) in the furrows and 50% WFP in the ridges, and the restrain effect decreased gradually with the depleting of soil moisture. The accumulated soil respiration calculated from the area ratio of the different parts in the furrows and ridges in the mulched field were both larger than in the non-mulched field during the growing season. However, this magnitude decreased with increasing precipitation over three experimental years. It was speculated that the effect of drip irrigation on the soil respiration was primarily on the ridges while the effect of precipitation mostly concentrated in the furrows and ridges in the non-mulched field because of the mulch barrier. Therefore, the precipitation accelerated more respiration in the mulched than in the non-mulched field. The difference in soil respiration between the mulched and non-mulched fields was observed to have a positive correlation with precipitation per the findings of other studies. In a humid climate with much more precipitation, soil respiration in the non-mulched field can also exceed that of the mulched field and explains why certain studies concluded that plastic mulch decreased soil respiration. The above results indicate that both irrigation and precipitation alter soil respiration and this effect can be modified by plastic mulch. Therefore, whether the PFM increases soil respiration compared to a non-mulched field largely depends on precipitation in the field.
**Keywords:** plastic film mulching; soil respiration; spatial variation; irrigation; precipitation

# 1. Introduction

Soil respiration, $R_s$, the flux of microbial- and plant-respired $CO_2$ from the soil surface to the atmosphere, represents the second largest $CO_2$ flux of the terrestrial



biosphere following gross primary productivity (GPP) and amounts to 10 times the
current rate of fossil-fuel combustion (Bond-Lamberty & Thomson, 2010, Davidson
*et al.*, 2006, Liu *et al.*, 2016a, Reichstein & Beer, 2008). Anthropogenic activities,
particularly agriculture expansion and cultivation changes, have brought significant
challenges to $CO_2$ emission control considering climate change over the twenty-first
century (Baker *et al.*, 2007). Further, the intensification of agriculture (the agricultural
Green Revolution) during the past five decades has been a driver of increasing the
seasonal amplitude of atmospheric $CO_2$ (Zeng *et al.*, 2014). The conversion of natural
to agricultural ecosystems causes a depletion of the soil organic carbon (SOC) pool by
as much as 60% in soils (Lal, 2004). Additionally, soil respiration in the cultivated
ecosystem is relatively larger than in natural ecosystems due to fertilization and
intensive cultivation (Buyanovsky *et al.*, 1987, Raich & Tufekciogul, 2000), such as
in arid regions where irrigation breaks the limits of soil moisture on soil respiration.
Since the 1950s, plastic film mulching (PFM) is one of the advanced agriculture
cultivation methods that have been widely applied around the world, e.g., in the
tropical USA, Europe, South Korea and China, as it can increase the soil temperature,
maintain soil moisture, promote seed germination, suppress weed growth and achieve
high yields (Anikwe *et al.*, 2007, Berger *et al.*, 2013). In 2014, approximately 19%
(25 million ha) of the total arable land (130 million ha) in China was cultivated using
PFM (Wang *et al.*, 2016). In the arid and semiarid parts of the Xinjiang Uygur
Autonomous Region in Northwest China, the PFM area has reached 1.2 million ha
within less than 20 years (Zhang *et al.*, 2014). Most of the fields have been converted
from the natural ecosystem for cotton production in the Xinjiang Uygur Autonomous
Region, the largest cotton production basin in China. The microclimate alterations,
which include the spatial and temporal albedo pattern, soil temperature, soil moisture,
and the caused change of crop growth, may affect both the heterotrophic and
autotrophic respirations in the PFM field. Further, the large-scale land use changes
may alter the regional climate, hydrologic cycle, and carbon cycle (Bonan, 2008, Cox
*et al.*, 2000, Li *et al.*, 2016). Therefore, detecting the altered environmental conditions
and $CO_2$ emissions in PFM field is crucial for the maintenance of regional and global
soil carbon balances in the situation of global climate change, which includes rising
atmospheric $CO_2$, increasing temperatures and shifting precipitation patterns.
The production of $CO_2$ in the soil is determined by root and microbial biomass,
the substrate supply, temperature and water conditions (Davidson *et al.*, 2006). Soil
respiration has an exponential increase with increasing temperature and the $Q_{10}$ value,
which is the factor by which respiration is multiplied when the temperature increases
by 10°C, is often used to define the sensitivity of soil respiration to temperature
(Davidson *et al.*, 2006, Fang & Moncrieff, 2001). However, $Q_{10}$ values also
incorporate the seasonal changes in the SWC, root biomass, litter inputs, microbial
populations and other seasonally fluctuating conditions and processes (Curiel Yuste *et*
*al.*, 2004). In suitable conditions, the soil moisture promotes soil respiration, which is
a benefit to root and microbe respiration. However, beyond this range, such as with
extremely low and high levels of soil moisture, the soil respiration is restrained by the
limited diffusion of the substrate and oxygen into water films and pore spaces,



respectively (Luo & Zhou, 2006). Furthermore, the soil moisture content and temperature are confounding rather than independent factors controlling the soil respiration. The effect of temperature and moisture on soil respiration are only regarding root and microbial responses to variations thereof throughout the soil (Davidson *et al.*, 1998).

The spatial and temporal pattern of soil temperature and moisture are modified significantly in a PFM field by the altering of the exchange of energy and water, and the momentum between the soil and atmosphere (Bonan, 2008). Plastic mulch hinders energy entry into the soil in daytime (Li *et al.*, 2016) with a high reflectance of radiation (Tarara, 2000) and preserves the heat flux at night, which results in a higher temperature under the mulch. The average mulched soil temperature within approximately 25 cm depth was 1-2 °C higher than the bare soil temperature (Gong *et al.*, 2015, Zhang *et al.*, 2011). The spatial pattern of soil temperatures in a PFM field was also affected by the crop growth and SWC (Zhang *et al.*, 2011). Soil moisture is preserved with PFM by reducing evaporation, forming a small water cycle beneath the plastic mulch (Yang *et al.*, 2016), and covering the soil with transparent polyethylene, which causes a significant increase in the soil moisture of the upper soil layer (Mahrer *et al.*, 1984). Combined with drip irrigation, a PFM approach can achieve better soil temperature and moisture conditions and obtain a higher yield and water use efficiency (Yaghi *et al.*, 2013). These environmental improvements promote microbial activity, which in turn enhances the mineralization rate of soil organic matter, thus providing readily available nutrients for plant growth, which simultaneously promotes the emission of greenhouse gases such as $CO_2$, $CH_4$ and $N_2O$ (Cuello *et al.*, 2015). However, Wang *et al.* (2016) note that PFM could also maintain the SOC level after six years of continuous cropping by balancing the increased SOC mineralization with increased root-derived carbon input, such as with straw incorporation in semiarid areas. Eddy flux experiments indicate that warmer and wetter soil stimulates GPP more than ecosystem respiration ($R_{eco}$) in a PFM field, which results in a higher net primary production (NPP) (Gong *et al.*, 2015).

In addition to soil temperature and moisture, the spatial heterogeneity of $CO_2$ concentrations and emissions are enhanced in a PMF field. Soil respiration involves two critical processes, which include the $CO_2$ produced in the soil by roots and microorganisms and that transferred through the soil profile to soil surface. The $CO_2$ concentration represents the production amount, and the emission represents the transfer amount (Luo & Zhou, 2006). Yu *et al.* (2016) showed that the $CO_2$ concentration in ridges was much larger than in the furrows. The $CO_2$ concentration in the ridges and furrows in a mulched field increased by 49% and 15%, respectively, compared to those in a non-mulched field. However, there was no difference in the $CO_2$ emission of the ridges in mulched and non-mulched fields. The main difference was in the furrows, where the $CO_2$ emission increased by 21%, and the cumulative $CO_2$ emission for the entire field increased by 8% in a mulched field relative to the non-mulched field. Li *et al.* (2011) also detected that $CO_2$ concentrations in the soil profile were higher in a mulched field than in the non-mulched field. However, the author found that the accumulated $CO_2$ flux in a mulched field decreased by 21%



relative to the non-mulched field. Further, the author argued that the plastic mulch
increased the soil-to-atmosphere pathway of $CO_2$ emission as most of the soil surface
(60%) was covered by mulch film, and the only pathways were furrows and small
plant holes. Therefore, the barrier of the plastic mulch would contain the $CO_2$
underneath, which would restrain $CO_2$ production and emission. Berger *et al.* (2013)
found extraordinarily low $N_2O$ fluxes from the plastic mulch and that $N_2O$ emission
from the plant hole were 68% that of the ridges in the non-mulch field. Nishimura *et*
*al.* (2012) revealed in a laboratory experiment that $N_2O$ gradually permeates the
plastic mulch and significantly emits from the furrows. These findings indicate that
the pathways for the $N_2O$ emission in a mulch field include the furrows between the
mulch (mf), the plant holes (mh) for crop germinating and the plastic mulch (mp) in
the ridges. However, the transport pathways for the $CO_2$ emission in PFM have not
yet been detected. Certain experiments simply interpreted soil respiration in the
furrows as the soil respiration of the whole field (Liu *et al.*, 2016b, Qian-Bing *et al.*,
2012), which may underestimate the results as the ridges emit more $CO_2$ than the
furrows (Yu *et al.*, 2016).
It is noteworthy that different climates may influence the effect of plastic mulch on
soil respiration. An example is that south of Xinjiang (precipitation 45.7 mm), PFM
increased the $CO_2$ emission (Yu *et al.*, 2016), while north of Xinjiang (precipitation
160 mm), the PFM decreased the $CO_2$ emission (Li *et al.*, 2011). In a semi-humid area
on the Loess Plateau of China (precipitation 500 mm), Xiang *et al.* (2014) found that
a plastic mulched treatment decreased the $CO_2$ emission by 39% because of the high
soil moisture and barrier of the plastic mulch. Still, in a temperate monsoon climate
(precipitation 1,954 mm) in Japan, Okuda *et al.* (2007) found that the annual $CO_2$
emission with the mulching decreased by nearly 40%. The author argued that the
high-water filled porosity might reduce the $CO_2$ emission. In a typical temperate
monsoon climate in South Korea (precipitation 1,440 mm), Berger *et al.* (2013) found
that PFM significantly decreased the $N_2O$ in a mulched field considering the
monitoring of plant holes and the plastic mulch. The above results indicate that in a
humid area with greater precipitation, the plastic mulch treatments all decreased the
soil respiration and the precipitation may affect the impacts of plastic mulch on soil
respiration.
Irrigation and precipitation are both crucial to soil respiration and the carbon cycle,
particularly in arid and semiarid regions. Irrigation is primarily applied to satisfy crop
requirements in arid and semiarid regions that have little precipitation. Precipitation
plays a dominant role in regulating the soil C balance in natural ecosystems in the arid
and semiarid regions (Lai *et al.*, 2013). Discrete precipitation pulses are important
triggers for the activity of plants and microbes and these factors combine to influence
the carbon balance (Huxman *et al.*, 2004). The effect of precipitation and irrigation on
soil respiration is related to the existing soil water condition, i.e., motivates soil
respiration in a dry soil and restrains soil respiration in moist soil (Dong, 2010). After
irrigation and precipitation, soil moisture undergoes a wetting-drying cycle that
affects the porosity of the soil and influences the activities of the root biomass and
microorganisms that control soil carbon dynamics (Yan *et al.*, 2014). The intensity





and amount of irrigation or precipitation both affect soil respiration. Certain studies indicate that soil respiration in a drip irrigation field was greater than in a flood irrigation field (Guo *et al.*, 2017, Qian-Bing *et al.*, 2012), and inter-annual variations in soil respiration were positively related to inter-annual fluctuations in precipitation (Liu *et al.*, 2009). The hydrological cycles after precipitation and irrigation are modified by plastic mulch application and may have a different influence on soil respiration. Precipitation cannot infiltrate ridges past the barrier of plastic mulch but can increase the runoff in furrows. Meanwhile, irrigation primarily infiltrates the soil in ridges in drip irrigation fields as the drip tapes are beneath the plastic mulch.

From the discussion above, the study of PFM on soil respiration is of great significance to regional and global agricultural carbon sequestration, and the spatial heterogeneity of the soil temperature, moisture and soil respiration are all enhanced in a PFM field. However, the effect of plastic mulch on soil respiration is still largely unclear, and this effect may be confounded by other factors such as irrigation and precipitation (Berger *et al.*, 2013, Li *et al.*, 2011, Yu *et al.*, 2016). In this study, we took advantage of the frequent irrigation and precipitation in the plastic- and non-mulched drip irrigation fields to discuss (1) how the spatial and temporal patterns of microclimate and soil respiration are affected by plastic mulch; (2) the effect of plastic mulch on soil respiration via its effect on soil temperature and moisture; and (3) the effect of irrigation and precipitation on soil respiration in mulched and non-mulched fields.

## 2. Materials and Methods

## 2.1 Site description

The field experimental site (86°12′ E, 41°36′ N; 886 ma.s.l.) is in an inland arid area, which is in one of the oases scattered on the alluvial plain of the Kaidu-Kongqi River (a tributary of the Tarim River) Basin, north of the Taklamakan Desert (Fig. 1) in the Xinjiang Uygur Autonomous Region in Northwest China. The region has a temperate continental climate, with a mean annual precipitation of 60 mm, mean annual temperature of 11.48°C, and mean annual potential evaporation of 2,788 mm as calculated by Φ20 pan. The annual sunshine duration is 3,036 hour, which is favorable for cotton growth. The experimental field has an area of 3.48 ha. The major soil texture in the field is silt loam, and the contents of sand, silt and clay are 32.8%, 62.4% and 4.8%, respectively. The soil bulk density of the experiment field is from $1.4 \text{ g cm}^{-3}$ to $1.64 \text{ g cm}^{-3}$ in the 1.5 m soil profile. The soil porosity is 0.42, which was directly determined in the laboratory using the known volume of undisturbed soil columns collected in the experimental field.

Cotton (Gossypium hirsutum L.) is usually sown in April and harvested from October to November. The planting style is "one film, one drip pipe beneath under the film and four rows of cotton above the film" (Fig. 1). The plastic film (0.008 mm





thick) is white and made of dense and airtight transparent polyethylene film. The
width of the film is 1.1 m, and the inter-film zone is 0.4 m. Before sowing, small
square holes (2 cm length) were made for germinating at 0.1 m intervals within a row
in the plastic film, and then seeds were placed into the holes, and finally, each hole
was covered with soil. The planting density was approximately 160,000 plants per ha.
The annual basic fertilizer before sowing included 173 kg ha$^{-1}$ of compound fertilizers
(14% N, 16% $P_2O_5$, and 15% $K_2O$), 518 kg ha$^{-1}$ of calcium superphosphate (18% N,
40% $P_2O_5$) and 288 kg ha$^{-1}$ of diammonium phosphate ($P_2O_5$>16%). Supplemental
fertilizers during the growth period included approximately 292 kg ha$^{-1}$ of urea (46%
N) and 586 kg ha$^{-1}$ of drip compound fertilizer (13% N, 18% $P_2O_5$, and 16% $K_2O$) and
foliar fertilizer ($P_2O_5$>52%, and $K_2O$>34%). Drip irrigation usually began on June 12
in the bud stages with an amount approximately 20-50 mm each time and
approximately 9-12 times per growing season. The annual irrigation amount was
approximately 400-600 mm.



## 235  2.2 Experimental design

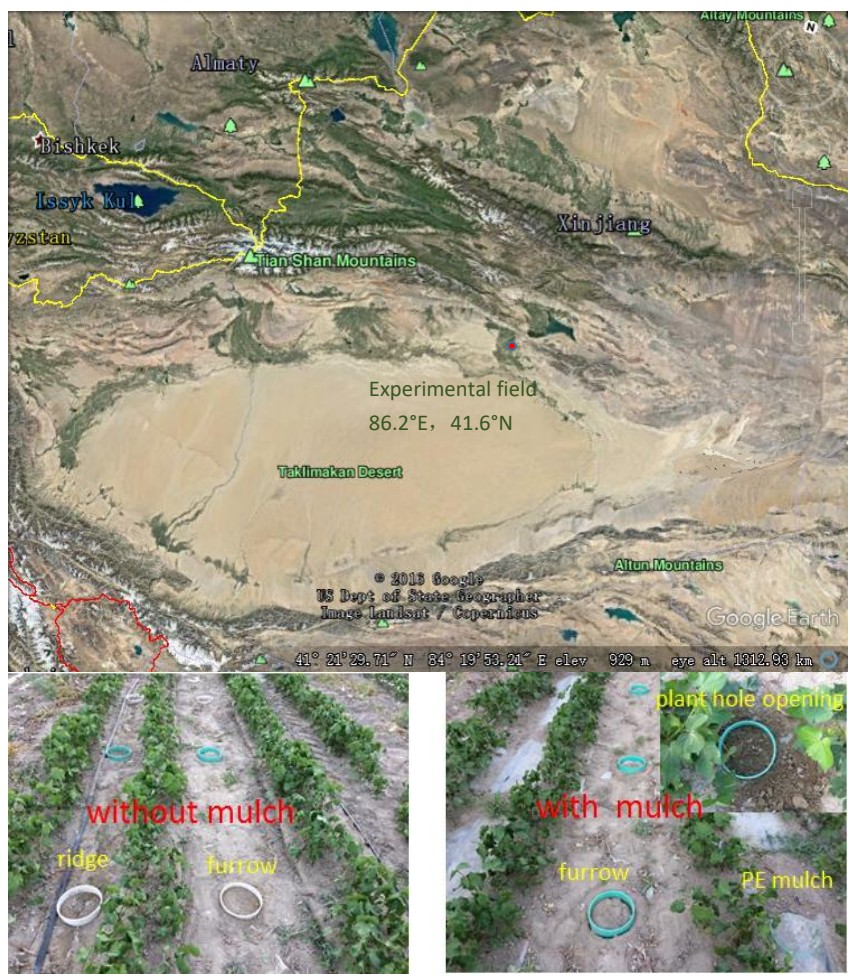


Fig. 1. Experimental site and experimental design. (a) Google map of the study area and the experimental site; (b)
Schematic drawing of the experimental design for the mulched and non-mulched fields.
The mulched and non-mulched treatments were arranged in a randomized block
design with three replicates in the same field and the same fertilization and irrigation
from the year 2014 to 2016. The plastic mulch was uncovered after the seed
germination in the non-mulched treatment to ensure the same seed germinating date
with the mulched field. The soil respiration was measured every two weeks during the
cotton-growing season with an LI-8100A (LI-COR, Inc., Lincoln, Nebraska). The
automated soil $CO_2$ flux system consisted of two parts, PVC collars (10 cm in
diameter and 5 cm in height) and a measuring chamber. The PVC collars were
inserted 2-3 cm into the soil by removing the small living plants and litter inside the
soil collars at least 1 day before the measurements. Data were recorded by the data





logger in the LI-8100. The soil respiration was measured in the furrows (nmf) and ridges (nmr) of the non-mulched treatment and the furrows (mf), ridges (mr), plant holes (mh), and plastic mulch (mp) of the mulched treatment. The measurements were performed every 2 hours during the day from 8:00 am to 24:00 pm. To measure the soil respiration on the soil surface without the plastic mulch covering, such as on the nmf and nmr in the non-mulched field and the mf in the mulched field, the PVC collars were inserted directly into the soil. Before measuring the $CO_2$ emission in the mp and mr, the plastic mulch was cut with a rectangle of 40 cm length and 30 cm width. Then, the collars were buried under the plastic mulch by compacting firmly with soil along the mulch edge. The $CO_2$ emissions in the mp were measured directly by placing the chamber on the covered collars. The $CO_2$ emission in the mr was measured by uncovering the plastic mulch. The $CO_2$ emission in the mh was measured by inserting collars into the soil, covering two plant holes along the direction of the mulch, and using scotch tape to seal the interspaces between the plastic mulch and collar.

The soil temperature and soil moisture at a depth of 5 cm were monitored adjacent to each PVC collar using the auxiliary sensors of the Li-8100, and concurrent with the soil $CO_2$ flux measurements. The drip irrigation amount was measured by water meters that were installed on the branch pipes of the drip irrigation system. The precipitation was measured by a tipping bucket rain gauge (model TE525MM, Campbell Scientific Inc., Logan, UT, USA), which was mounted 0.7 m above the ground.

## 2.3 Data calculation and analysis method

The soil respiration of the different parts at a particular time of a day was the average of three replicates. The daily mean soil respiration was calculated using the average of the soil respirations measured at various times in a day. The soil respirations in the mulched and non-mulched fields were calculated relying on the area ratio of the various parts in the field:

$$R_{mr} = R_{mh} * A_{mh} + R_{mp} * A_{mp}$$
$$R_{sm} = R_{mf} * A_{mf} + R_{mr} * A_{mr}$$
$$R_{snm} = R_{nmf} * A_{nmf} + R_{nmr} * A_{nmr} \qquad (1)$$

where $R_{sm}$ and $R_{snm}$ are the soil respirations in the mulched and non-mulched fields, respectively. The symbols of ($R_{mh}$) and ($R_{mp}$) are the soil respirations in the plant holes and plastic mulch, which constitute the soil respiration in mr ($R_{mr}$). The symbols of $R_{mr}$, $R_{mf}$, $R_{nmr}$, and $R_{nmf}$ are the soil respirations in the furrows and ridges in the mulched and non-mulched fields, respectively. Replacing the initial letter $R$ with $A$ means the area ratio of the different parts. The accumulated soil respiration in the ridges and furrows during the growing season were estimated by summing the products of the soil $CO_2$ flux and the number of days between sampling times.

The regression and smoothness of the soil respiration with soil temperature and SWC were analyzed using SPSS software. The Van't Hoff equation was used to





express the relationship of the soil respiration with soil temperature (Hoff, 1898):
$$R_s = Ae^{bT} \tag{2}$$

where $R_s$ is the soil respiration, $T$ is the soil temperature, $A$ is the intercept of the soil
respiration when the temperature is 0°C (i.e., reference soil respiration). Moreover, b
represents the temperature sensitivity of the soil respiration. The $Q_{10}$ value, which
describes the change in soil respiration over a 10-°C increase in the soil temperature, is
calculated as
$$Q_{10} = e^{10b} \tag{3}$$

298   Considering a lower and higher SWC both restrain the soil respiration, we use a
quadratic equation to simulate the effect of soil moisture on soil respiration (Davidson
*et al.*, 1998):
$$R_s = aV^2 + bV + c \tag{4}$$

where $V$ is the soil water content and a, b and c are fitted constants.





## 3. Results

## 3.1 Field microclimate and crop growth

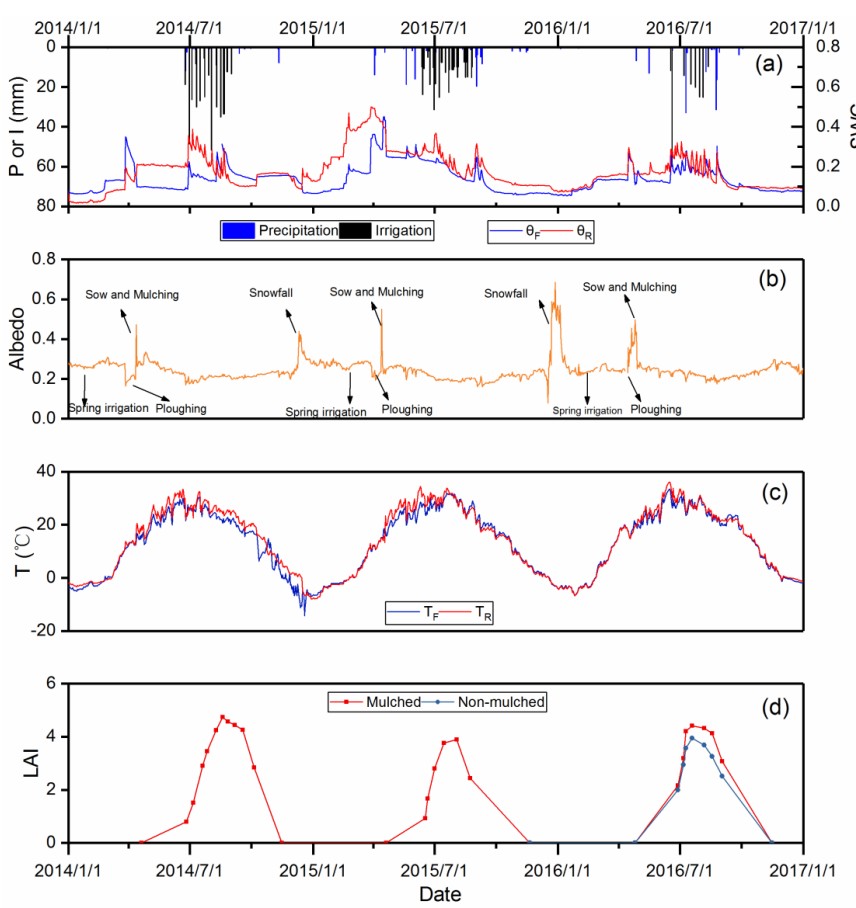

Fig. 2 Microclimate affected by plastic mulch, irrigation and precipitation. (a) The SWC in the ridges ($\theta_R$) and furrows ($\theta_F$) affected by irrigation and precipitation. (b) The albedo in the mulched field. (c) The soil temperature in the furrows ($T_F$) and ridges ($T_R$) in the mulched field; (d) The leaf area index (LAI) in the mulched and non-mulched fields (LAI in the non-mulched field was only measured in 2016 to compare to that in the non-mulched field).

The plastic mulch altered all the field microclimate aspects such as the albedo, and soil conditions such as the soil temperature and moisture, and crop growth conditions. There were two snowfalls during January 2015 and January 2016 that resulted in much higher albedo, which was beyond 0.4. The spring irrigation used a month before sowing to apply the germinating water and washing soil salt in early March increased the albedo. Tillage significantly decreased the albedo several days before mulching on





April 20. After the plastic mulch covering in April, the surface albedo had a sudden
rise, and then, slowly decreased with the increase of the crop canopy and applied
irrigation. The albedo reached the minimum value with the highest value of LAI at the
bud stage during August, and then, increased very slowly with leaf fall.
The soil temperature was highly correlated with radiation over a growing season,
and it was affected by the plastic mulching and irrigation. The soil temperature in the
ridges with mulch covering was significantly higher than in the furrows without
mulch covering. However, in the later growth stages, the soil temperature in the
furrows exceeded that in the ridges as the crop canopy and irrigation increased. The
soil temperature decreased significantly after irrigation and two heavy rainfall events
during 2016, and the variation in soil temperatures during the growing season was as
drastic as the effect of frequent irrigation.
The soil moisture varied in response to irrigation and precipitation, and the greater
the irrigation and precipitation, the more drastic the variation. The soil moisture in the
ridges was mostly larger than in the furrows with the effect of frequent drip irrigation.
However, after heavy rainfall, the soil moisture in the furrow exceeded even that in
the ridge, i.e., during the two heavy rainfall events on July 10 and August 24 of 2016,
which were 36.8 mm and 47.9 mm, respectively. Inter-annually, the soil moisture in
the furrows during 2016 was larger than in 2014 and 2015 because of the greater
precipitation during 2016, and the soil moisture in the ridges during 2016 was lower
than that during 2014 and 2015 because of the smaller amount of irrigation.
The plant phenology and LAI showed the growing-dying cycle varying with
temperature and radiation over the seasons. The LAI started increasing with seed
germination, reached its maximum value at the bud stage during August, and then,
decreased with the leaf falling. The LAI in the mulched field was significantly larger
than in the non-mulched field during 2016, particularly in the vigorous growth stages.
Inter-annually, the LAI during 2016 was the greatest and that during 2015 was
smallest.





## 3.2 Seasonal and spatial variations in soil respiration

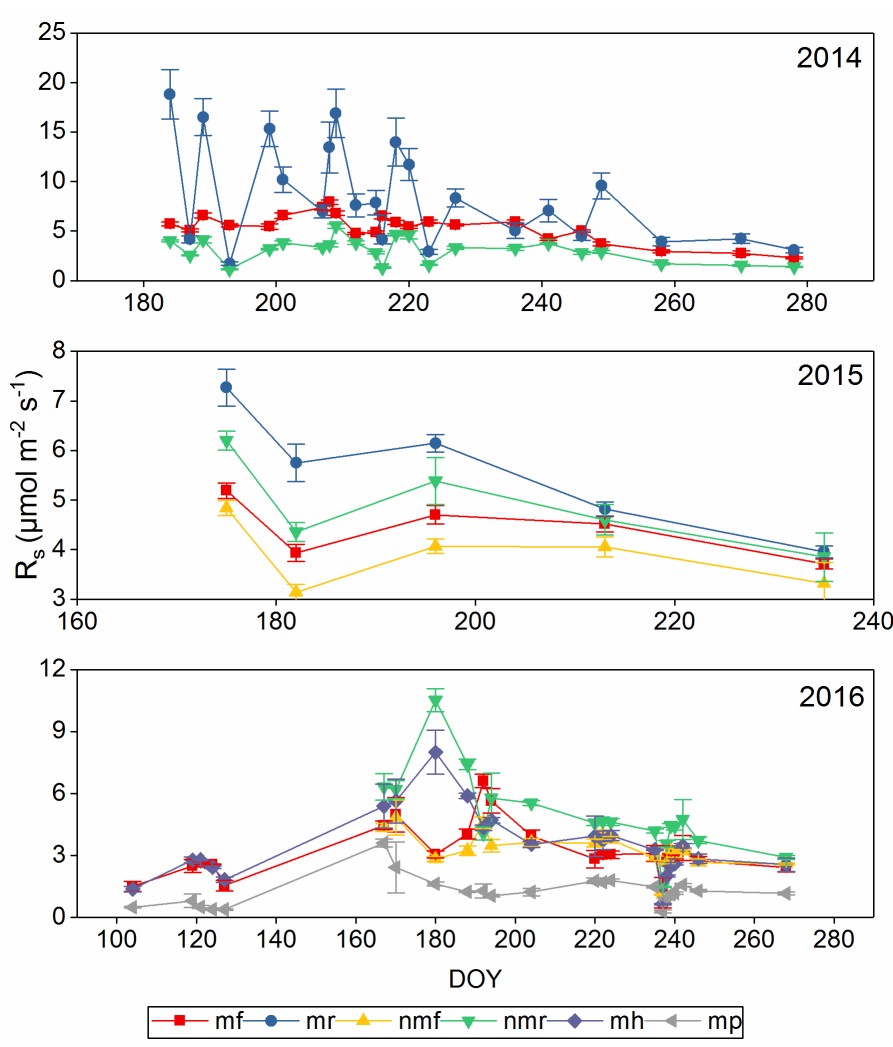

Fig. 3 Seasonal variations in soil respiration in different parts of the mulched and non-mulched fields over the
three years. Data represent means over a day ± SD of three replicates.

The seasonal variations in the soil respiration over three years were approximately
consistent with the radiation, temperature and LAI. In the non-growing season, the
soil respiration was very low from October to April of the next year, i.e.,
approximately 1 to 2 $\mu$mol m$^{-2}$ s$^{-1}$, and reached a peak value in the middle of July
during summer, approximately 6 to 8 $\mu$mol m$^{-2}$ s$^{-1}$. After tillage in April of 2016, the
soil respiration was significant and then had a rapid decline with the plastic mulching.
The inter-annual variation in the soil respiration during the three years was not very
significant. The highest values during 2014 to 2016 were approximately 8 $\mu$mol m$^{-2}$



$s^{-1}$, 6 µmol $m^{-2}$ $s^{-1}$ and 7 µmol $m^{-2}$ $s^{-1}$, respectively, which was consistent with the
highest LAI values of approximately 4.2, 3.8 and 4.2, respectively. The seasonal
variations in the soil respiration were altered by both the irrigation and precipitation.
The irrigation obviously restrained the soil respiration during 2014, with the soil
respiration significantly decreasing to an extremely low value right after irrigation,
and then, rising with the evapotranspiration of soil moisture. The soil respirations in
the mh, mp and ridges in the nmr during 2016 almost had the same variation of
response to irrigation. Meanwhile, the soil respiration in the furrows in the mf and the
nmf during 2016 had the same variation because they were both directly affected by
precipitation and indirectly affected by irrigation. The precipitation significantly
restrained the soil respiration of all parts in the mulched and non-mulched fields after
a large rainfall at the day of the year (DOY) 238 in 2016.
The spatial heterogeneity was more enhanced in the mulched field than in the
non-mulched field. In the non-mulched field, the soil respiration in the nmr with the
higher SWC was always larger than that in the nmf with a lower SWC. Meanwhile, in
the mulched field, the soil respiration in the mh exceeded that in the mf, except after
the 36.8 mm rainfall in DOY 199. The soil respiration in the mp was lower at the
beginning, approximately 1 µmol $m^{-2}$ $s^{-1}$. However, it rose to approximately 2.75
µmol $m^{-2}$ $s^{-1}$ by the bud stage. The soil respiration in the mr measured by uncovering
the plastic mulch during 2014 was extremely high, approximately 15 µmol $m^{-2}$.

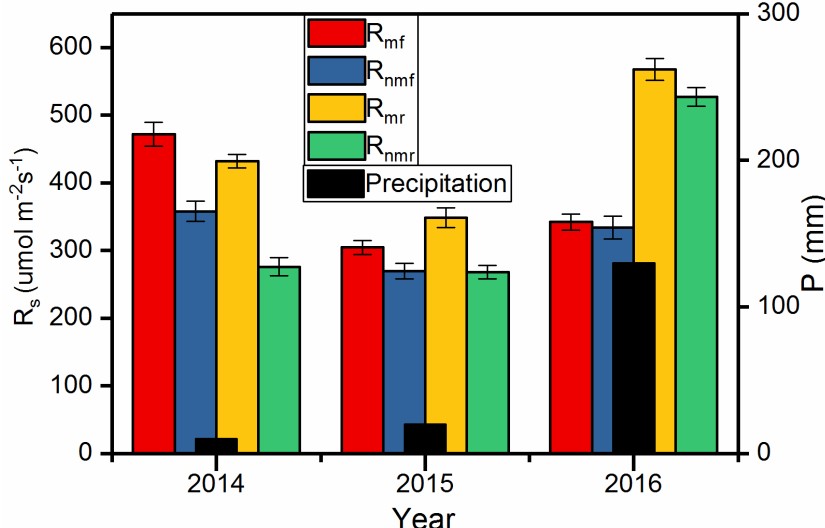


Fig. 4 The seasonal accumulative soil respiration affected by precipitation. The data represent the seasonal
accumulated soil respiration in the furrows ($R_{mf}$) and ridges ($R_{mr}$) of the mulched field and the furrows ($R_{nmf}$) and
ridges ($R_{nmr}$) of the non-mulched field, and the precipitation during the growing season over three years. The error
bars represent standard deviations.
The accumulated soil respirations calculated per the area ratio of different parts in
the ridges and furrows in the mulched field were both larger than those in the
non-mulched field. The average accumulated soil respiration was 428.91 µmol $m^{-2}$ $s^{-1}$



in the mulched field and 347.13 μmol m$^{-2}$ s$^{-1}$ in the non-mulched field during the
growing season over three years. However, the differences in the soil respiration in
the furrows were all smaller than in the ridges and the differences in the ridges and
furrows between the mulched and non-mulched fields all decreased from the year
2014 to 2016. It is noteworthy that the amount of precipitation increased from 2014 to
2016, which may have had some influence on the different soil respirations in the
mulched and non-mulched fields.



### 3.3 Soil temperature and soil respiration

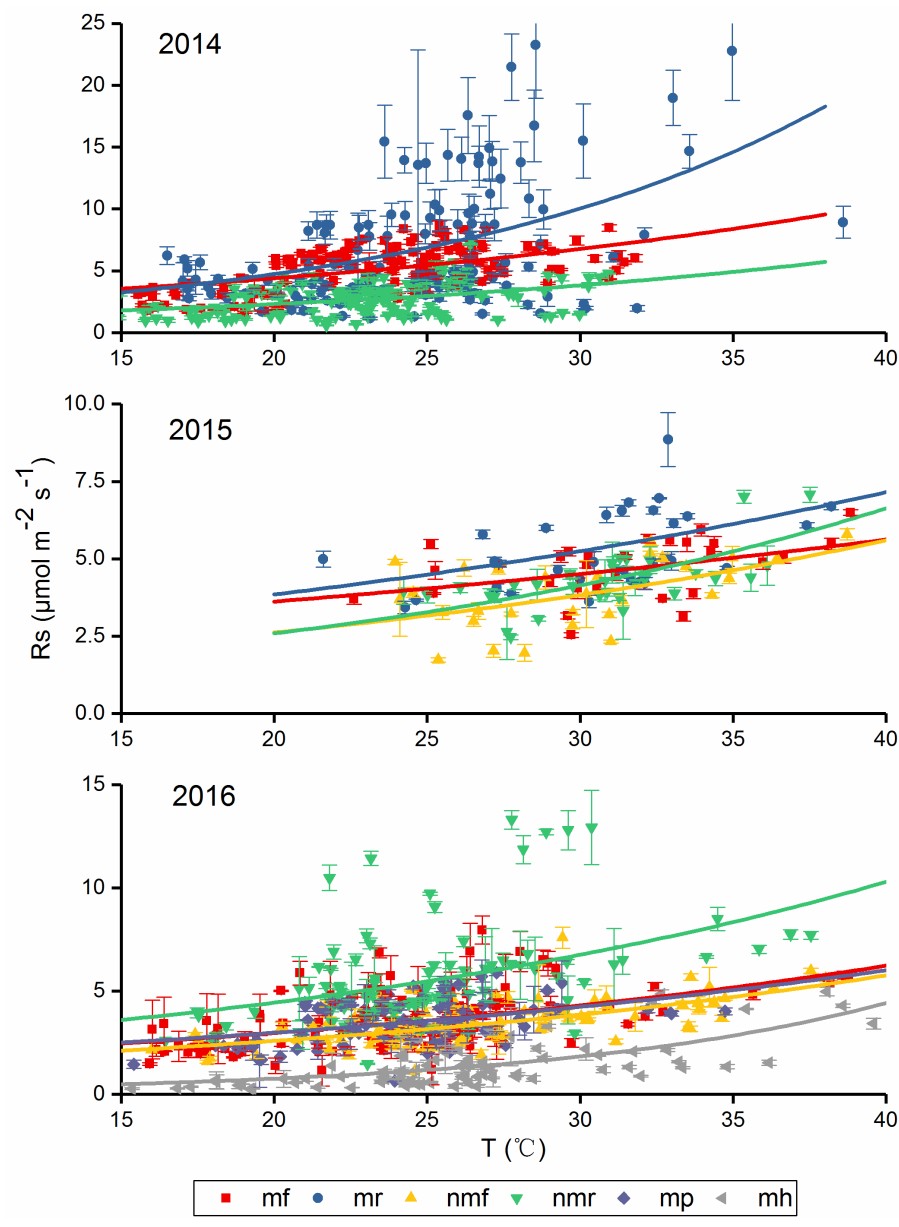

Fig. 5 The relationship between soil respiration and soil temperature. The data represent means ± SD of three replicates. The smooth lines of the different parts were fitted with Equation 1.

The soil respirations had distinct seasonal variations that were determined primarily by the radiation, temperature and phonology although they were also

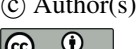



frequently affected by irrigation (Fig. 3). The soil respiration in different parts of the
mulched and non-mulched fields all increased with temperature and can be expressed
using exponential equations (Fig. 5). However, their correlation $R^2$ and $Q_{10}$ values
were very different and weakened by the extreme variations in the soil moisture with
an $R^2$ smaller than 0.5 and $Q_{10}$ values lower than 2.0 (Table 1). The reference soil
respiration ($A$ in Equation 1) during 2015 was larger than during 2014 and 2016
because the observation time was limited and the temperature variation range was
small. The correlations of soil respiration in the furrows were better than those in
ridges, while the $Q_{10}$ values in the furrows were much lower than those in the ridges.



Table 1 Exponential equations of the soil respiration with soil temperature

| Year | Parameters | mf | mr | nmf | nmr | mh | mp |
|------|-----------|------|------|------|------|------|------|
| **2014** | a | 1.87 | 1.06 | | 0.86 | | |
| | b | 0.04 | 0.08 | | 0.05 | | |
| | $Q_{10}$ | 1.54 | 2.12 | | 1.65 | | |
| | $R^2$ | 0.29 | 0.18 | | 0.18 | | |
| **2015** | a | 2.33 | 2.07 | 1.23 | 1.01 | | |
| | b | 0.02 | 0.03 | 0.04 | 0.05 | | |
| | $Q_{10}$ | 1.25 | 1.36 | 1.46 | 1.60 | | |
| | $R^2$ | 0.18 | 0.27 | 0.27 | 0.43 | | |
| **2016** | a | 1.42 | | 1.16 | 1.92 | 1.48 | 0.13 |
| | b | 0.04 | | 0.04 | 0.04 | 0.04 | 0.09 |
| | $Q_{10}$ | 1.45 | | 1.49 | 1.52 | 1.42 | 2.41 |
| | $R^2$ | 0.23 | | 0.39 | 0.20 | 0.18 | 0.44 |



## 3.4 Irrigation and soil respiration


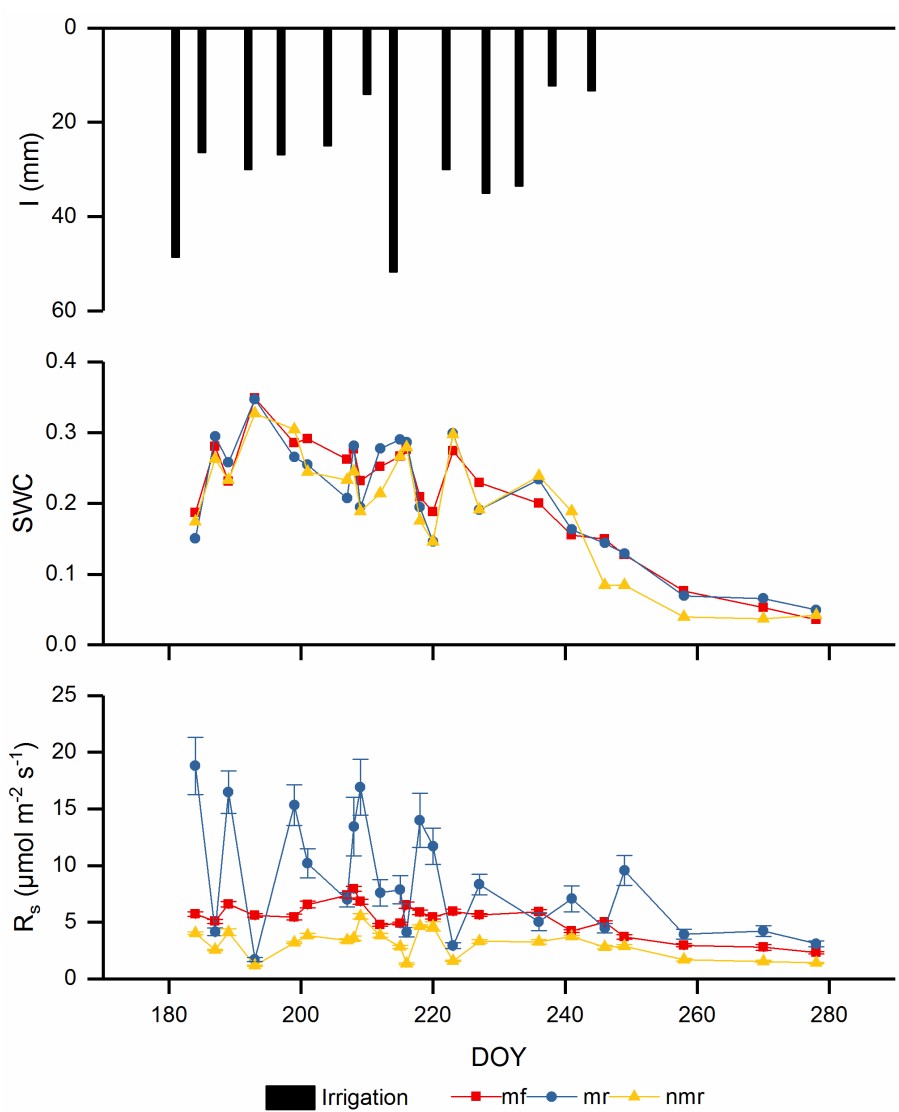


Irrigation ■ mf ● mr ▲ nmr

Fig. 6 The responses in soil moisture and soil respiration of the different parts to irrigation in the mulched and
non-mulched fields during the year 2014.
The soil moisture and respiration were significantly dynamic and fluctuated during
the growing season under the influence of frequent irrigation. However, the responses
to irrigation varied as the soil moisture and respiration increased and decreased,
respectively, after irrigation. Therefore, more irrigation led to a larger variation in the
soil moisture and respiration. This finding indicates that after irrigation, the soil
moisture increased but that the soil respiration was restrained. Variations in the soil



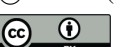

moisture and respiration in mr and nmr were more drastic than in mf. The soil
moisture and respiration in the mr and nmr had the same variations as these factors
both responded to irrigation immediately. Meanwhile, the soil moisture and
respiration in mf were slower to respond to the irrigation. As the evaporation in the
nmr was drastic in the arid area without the protection of the plastic mulch, the soil
moisture in the nmr was always lower than in the mf over time, except for
immediately after irrigation. This factor caused the soil respiration in the nmr to
always be lower than in the mf.

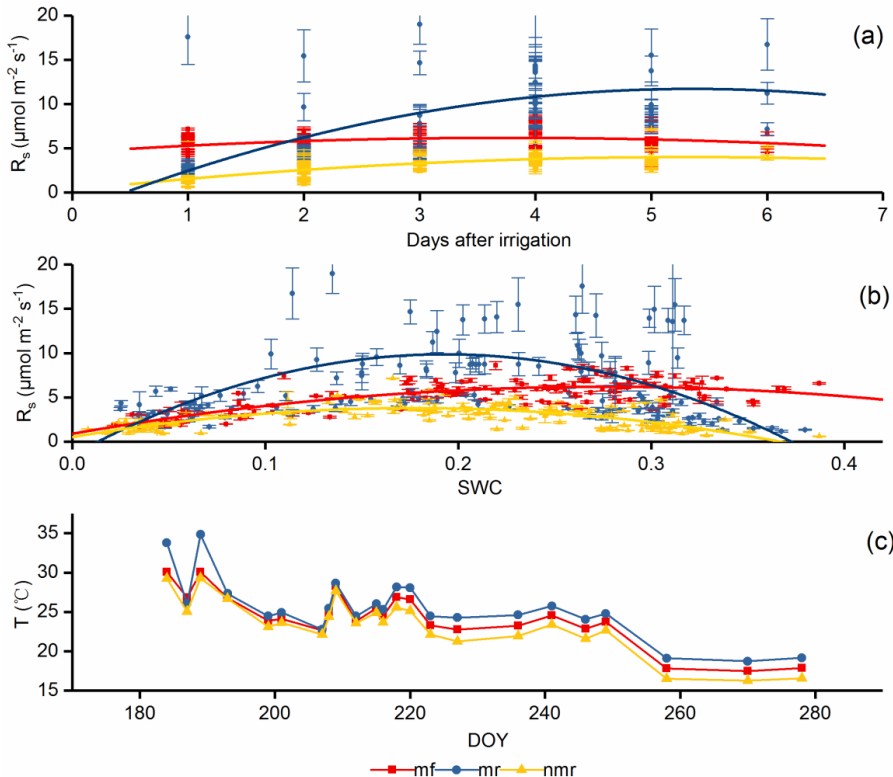

Fig. 7 The soil respiration affected by irrigation. The data represent the average of three duplicates; the error bar
represents standard deviation. The fitted lines were used with the binomial equation. (a) Variations in the soil
respiration within days after irrigation. (b) The relationship between the soil respiration and soil moisture. (c) The
soil temperature affected by irrigation.
The effect of irrigation on the soil respiration was presented by the soil respiration
relationship and days after irrigation with an irrigation cycle of approximately 6 days.
The soil respirations were extremely low after irrigation in the mr and nmr, and then,
recovered slowly within days after irrigation. Meanwhile, as in the mf, the soil
respiration was almost unaffected by irrigation and only had a litter rise on the fourth
day (Fig. 7a). The three parts reached the maximum values in 4 days and began to
decrease with the decrease in the soil moisture. The relationship between soil the



respiration and soil moisture can be expressed in the form of a binomial equation.
Before irrigation, the soil respiration was extremely low in the drier soil, and then it
increased with the rising soil moisture. However, the soil respiration began to decline
when it reached a threshold. The soil moisture threshold that caused the decline of the
soil respiration was approximately 0.25 in the mf and approximately 0.2 in the mr and
nmr (Fig. 7b). Moreover, these soil moisture thresholds were approximately 60% and
50% of the water-filled pore space (WFP), respectively. The soil temperatures in the
nmr and sometimes in the mr were smaller than in the mf due to the effect of
irrigation. The restrain threshold in the mf was smaller than in the mr, which could be
because in the ridges, the irrigation not only increased the soil moisture but also
decreased the soil temperature, i.e., reducing soil respiration (Fig. 7c).





### 3.5 Precipitation and soil respiration

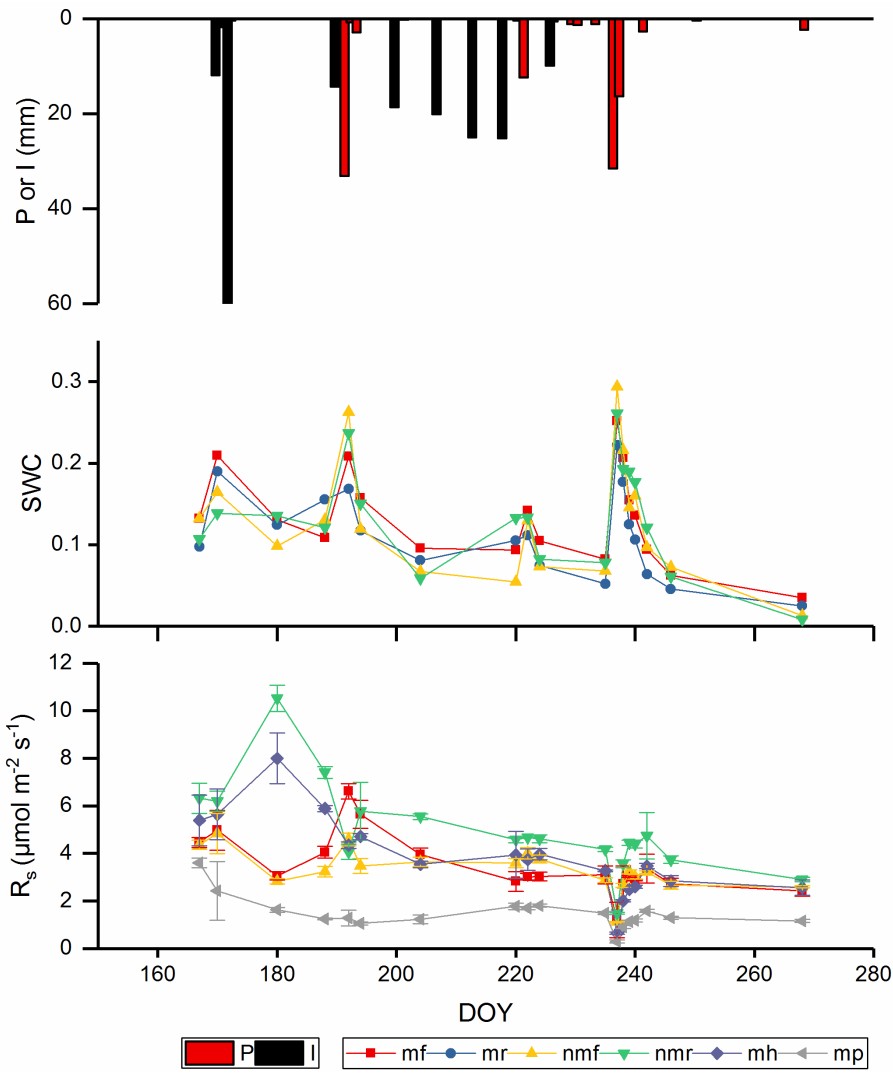

Fig. 8 The response of the soil moisture and soil respiration to precipitation and irrigation during 2016.

In 2016, there were three big rainfalls of 36.8 mm, 12.8 mm, and 48 mm in the DOY 192, 222, and 237, respectively. The soil moisture increased significantly after the 36.8 mm and 48 mm rainfalls but only slightly after the 12 mm rainfall. The soil moisture in the furrows was greater than in the ridges, and the soil moisture in the nmr was greater than in the mr, sometimes even larger than in the mf after precipitation. The soil respiration in the nmr was always greater than in the mp and mf, which was different during 2014 and 2015. Different amounts of precipitation had various effects on the soil moisture and respiration. The 12 mm precipitation had little effect on the



soil moisture and respiration. The 36.8 mm precipitation increased the soil moisture in
the mf, nmf and nmr, but had little effect on the soil moisture under the plastic mulch
(mr) because of the plastic mulch barrier. This precipitation restrained the soil
respiration in the mr and mh but motivated the soil respiration in the mf and nmf. The
48 mm precipitation increased the soil moisture in all the parts except for the mr, and
restrained the soil respiration in all the parts of the mulched and non-mulched fields.

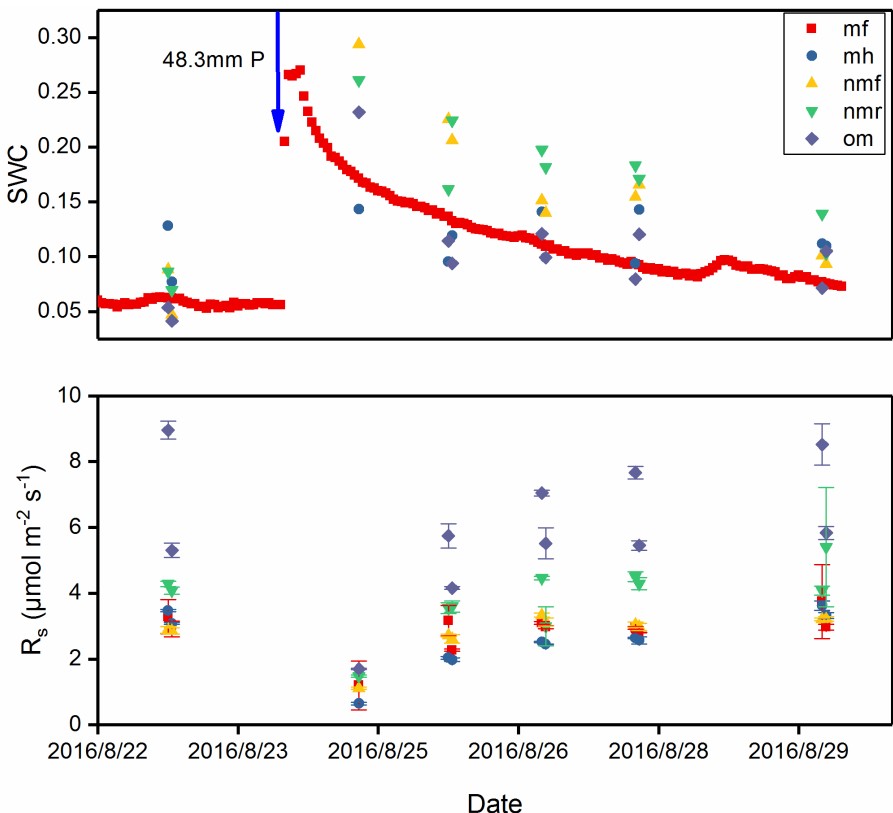


Fig. 9 Variations in the soil moisture and respiration in a wetting-drying cycle after a big rainfall (om means
opening mulch and is the soil respiration in the ridges after uncovering the plastic mulch for 24 hours).
The effect of precipitation on the soil respiration in a wetting-drying cycle was
studied carefully before and after a substantial rainfall of approximately 48 mm on
August 24, 2016. The soil respiration was significantly restrained by the high SWC
both in the furrows and ridges in the mulched and non-mulched fields. The restrain
was relieved by the evapotranspiration of the soil moisture. Soil respirations in the
different parts were all restrained although the SWCs were very different in the
various parts and the lowest SWC was 0.15 in the ridges under mulch. This finding
means that the soil respirations were all restrained when the SWC was greater than
0.15, which was less than the threshold value affected by irrigation. After rainfall, the
soil moisture in all the parts rose rapidly except in the ridges under the mulch due to
the barrier of plastic mulch and canopy interception. The soil moisture after rainfalls



was nmf>mf>nmr>mr, but the soil respiration after rainfalls was nmr>mf>nmf, which
means that precipitation primarily affected soil moisture in the furrows and ridges in
the non-mulched field, and a higher soil moisture restrained more soil respiration. The
soil respiration in the mh did not change much as the soil moisture in the ridges under
the mulch was nearly unaffected by precipitation. Several days after restrain, the
weakened soil respiration in the nmr was significantly larger than in the nmf and mf
because precipitation supplies more water to the ridges in the non-mulched field than
in the mulched field. It is noteworthy that it took approximately one day for the soil
respiration to reach a normal level after precipitation, which was much shorter than
the effect of irrigation. The soil respiration in the om, which was that under the mulch
measured by uncovering the plastic mulch for more than 24 hours, was significantly
greater than in the other parts, though it was also restrained.
To verify that different climate patterns may have different effects on the soil
respiration in the mulched and non-mulched fields, other studies regarding
comparative experiments in mulched and non-mulched fields were conducted to study
the effect of precipitation on the differences in soil respiration (dF) in mulched and
non-mulched fields (Fig. 10). Other studies included an arid area ($P$ 45.7 mm) south
of Xinjiang in China (Liu *et al.*, 2002), a semiarid area ($P$ 160 mm) north of Xinjiang
in China (Li *et al.*, 2011), a semi-humid area ($P$ 566.8 mm) on the Loess Plateau of
China (Xiang *et al.*, 2014) and an area in a temperate monsoon climate ($P$ 1,954 mm)
in Japan (Okuda *et al.*, 2007). Our experiments were added to these analyses, and the
climate in our research was an arid area south of Xinjiang with an annual precipitation
of 60 mm, except for 2016, which was a rainy year with 130 mm precipitation. Here,
dF means the difference in the soil respirations between the mulched and
non-mulched fields. The dF was found to have a linear relationship with the
precipitation amount. This factor increased with precipitation, and at 200 mm
precipitation, the soil respirations in the mulched and non-mulched fields were equal.
At precipitation outside 200 mm, the soil respiration was lower in the mulched than in
the non-mulched fields, e.g., 685 mm precipitation is a semi-humid area, and 2,000
mm is a temperate monsoon (Okuda *et al.*, 2007, Xiang *et al.*, 2014).





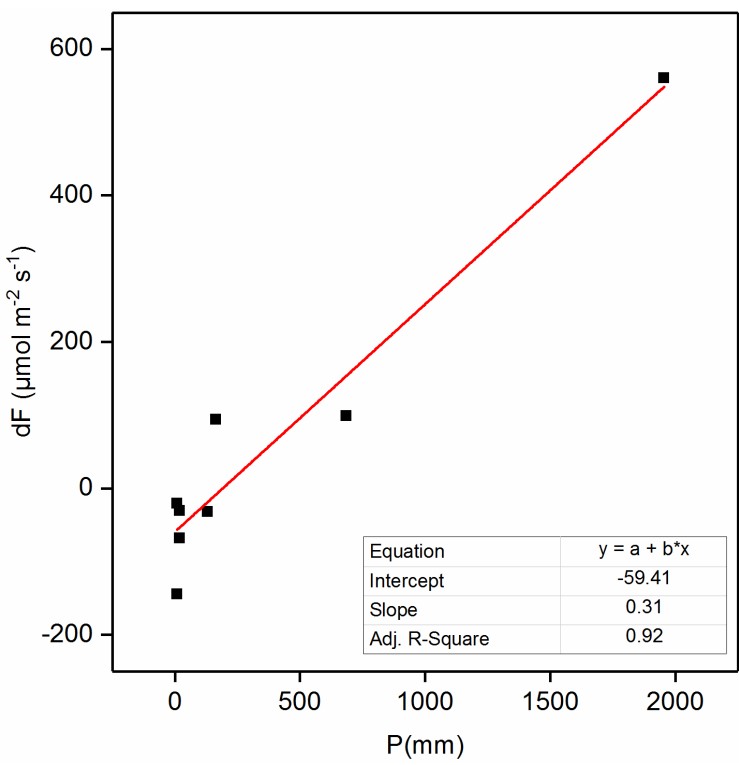


Fig. 10 The relationship of the difference in soil respirations in the mulched and non-mulched fields with
precipitation.
## 4. Discussion
### 4.1 Effect of plastic mulch on soil respiration
The production and transfer of $CO_2$ in the soil are both affected by the plastic
mulch. The production of $CO_2$ in the soil is determined by the root and microbial
biomass, substrate supply, temperature and desiccation stress (Davidson *et al.*, 2006).
The soil temperature, soil moisture and crop growth are all improved in the mulched
field relative to in the non-mulched field. Plastic mulch preserves heat and energy
transfer and soil moisture. Irrigation water is fully utilized as evaporation is prohibited
and transpiration increases due to theaccelerated crop growth absorbing more water
through the roots (Tian *et al.*, 2016, Yang *et al.*, 2016). Improved crop growth
produces more root biomass and litter fall in a mulched field, which will promote root
respiration and litter fall decomposition. Moreover, improved soil temperature and
soil moisture would promote the activities of the roots and microorganisms. Our
results indicate that the soil respiration in the ridges of the mulched field (mr) as





measured by uncovering the plastic mulch was much greater than in the furrows (mf). This finding indicates that indeed much $CO_2$ gathers beneath the plastic mulch because of the plastic mulch barrier. The soil respiration in the ridges after uncovering the mulch for 24 hours (om) (Fig. 9) was also prominently greater than in the ridges of the non-mulched field (nmr). This finding indicates that the suitable temperature and moisture environment in the ridges indeed produce more $CO_2$ in the mulched field than in the non-mulched field. Yu *et al.* (2016) also found that $CO_2$ concentrations in the ridges and furrows increased by 49% and 15%, respectively, in the soil of 0-40 cm.

Some researchers argued that the high concentration of $CO_2$ under the plastic mulch would restrain $CO_2$ production in the soil. However, as we know, the soil respiration is the by product for the survival of microorganisms and the root, and so the concentration of $CO_2$ in deeper soil is much higher than at the surface layer (Luo & Zhou, 2006). The $CO_2$ can emit via the horizontal diffusion of $CO_2$ from the ridge soil covered with mulch to the adjacent furrow (Nishimura *et al.*, 2012) and also through the plant holes and plastic mulch. Our experiment indicates that the plant holes emit more $CO_2$ than the furrows (Fig. 3), although the plant holes are soil-covered and only occupy small areas of mulch. However, the root biomass primarily concentrates around the plant holes, which can produce more root respiration. The plastic mulch itself can also emit up to 2.75 $\mu$mol $m^{-2}$ $s^{-1}$ $CO_2$. Considering that the plastic mulch occupies most of the ridge area, it is an important pathway for $CO_2$ emission in the mulched field. The emission rate of the plastic mulch correlates with the qualities of the plastic mulch, such as its thickness, texture and color. For example, a thick black PE mulch has an extraordinarily low $N_2O$ emission (Berger *et al.*, 2013), while high $N_2O$ is emitted from a polyethylene film only 0.02 mm thick (Nishimura *et al.*, 2012). Liu *et al.* (2016) also reported that the transparent plastic film emits more $CO_2$ than the black plastic mulch. The local farmers widely use the clear polyvinyl chloride (PVC) film with a thickness of only 0.008 mm as it can save on costs and absorb little but transmit up to 90% of solar radiation. This film has a relatively high diffusion for greenhouse gases. Therefore, the plant holes, furrows and plastic mulch are primarily responsible for $CO_2$ emissions in a mulched field, while only the furrows and ridges are responsible for $CO_2$ emissions in the non-mulched field.(Bi *et al.*, 2007)

Our results indicate that the plastic mulch accelerates soil respiration. The accumulated soil respirations in the ridges and furrows of the mulched field were greater than in the non-mulched field when considering the plant holes, furrows and plastic mulch. This result is a little different from that of Yu *et al.* (2016), who reported that soil respirations between the ridges were similar, while only soil respirations in the furrows in the mulched field were greater than in the non-mulched field. Liu *et al.*, (2016) also reported that transparent and black plastic films emit more $CO_2$ in the furrows, and (Cuello *et al.*, 2015) found that plastic film significantly increased the $CH_4$ and $N_2O$ greenhouse gas emissions.



## 4.2 Effect of irrigation on soil respiration

The soil respiration was strongly dynamic and fluctuated due to the drastic variations in the soil moisture because of the effect of frequent irrigation in the field (Fig. 6). In the wetting-drying cycle, the SWC reached a high lever right after irrigation, which restrained the soil respiration to an extremely low level. Moreover, in the subsequent period, the SWC was gradually depleted as water evaporated from the soil surface and was transported from the foliage canopy, which gradually increased the soil respiration. Soil respiration right after a big precipitation was also restrained significantly (Fig. 9). In the agriculture field, SWC was maintained at a relatively high level, i.e., greater than 20% in our experiment. Because the plastic mulch can preserve soil moisture by preventing evaporation, soil respiration was restrained after each irrigation. The frequency and amount of irrigation both affected the soil respiration by affecting the SWC. Xu *et al.* (2004) also found that the magnitude of the respiratory pulses was inversely related to its pre-rain value, and the decay of the respiratory pulses after the rain event was a function of the rainfall amount. In certain precipitation manipulating experiments, adding water significantly increased the soil respiration during a drought period (Liu *et al.*, 2002), but had no effect on soil respiration when the soil moisture was already relatively high (Lai *et al.*, 2013). This finding indicates that the effect of adding water such as through irrigation or precipitation manipulating experiments on soil respiration is related to the existing SWC, and it could result in soil respiration in dry soil and restrain soil respiration in a soil with a high-water content (Dong, 2010).

Our results indicate that both low and high SWC restrains soil respiration (Fig. 7b). The high-water-content restrain was caused by post irrigation during the growing season, while most of the low moisture content was because of no irrigation after the growing season. The soil moisture affected the soil respiration directly via the physiological processes of roots and microorganisms, and indirectly via diffusion of the substrate and $O_2$ (Luo & Zhou, 2006, Moyano *et al.*, 2012). Low water content affects the diffusion of soluble substrates, while a high-water content affects the diffusion and availability of oxygen (Davidson *et al.*, 2006, Linn & Doran, 1984). To satisfy crop water requirements and achieve high yield, frequent irrigation was applied in the field, i.e., the local irrigation was performed 13 times at an interval of 5-7 days. The relatively steady water conditions rendered the soil respiration always higher than that of natural ecosystems, particularly in the arid areas.

The sensitivity of the soil respiration to temperature was weakened by irrigation (Table 1). The correlation of soil respiration with the soil temperature in different parts of the mulched and non-mulched fields was not so good. Moreover, the $R^2$ was smaller than 0.5, particularly for the soil respiration in ridges. The $Q_{10}$ values were smaller than 2.0 except for in the plant holes, and $Q_{10}$ values in the furrows with a low SWC were smaller than in the ridges. This finding means that the soil respiration was less sensitive to temperature changes in the water-limited soils, which leads to lower $Q_{10}$ values (Liu *et al.*, 2016a). It was noteworthy that the threshold values of the SWC





restraining soil respiration were different in the mulch and non-mulched fields In the
furrows without plastic mulch, the value was 60% of the WFP, which is equivalent to
the former experimental results (Linn & Doran, 1984). However, in the ridges with
plastic mulching, the threshold value was only 50% of the WFP (Fig. 6). This finding
may be because the soil respiration was more sensitive to soil moisture in a lower
temperature range because the soil moisture in ridges was higher than that in the
furrow, while the temperatures were lower than in the furrow. Therefore, the effect of
soil moisture on the soil respiration was confounded with soil temperature (Davidson
*et al.*, 1998).

## 4.3 Effect of precipitation on soil respiration

From the 48 mm precipitation event, we can see the effect of the soil moisture on
soil respiration in the wetting-drying cycle. An extremely high SWC right after
precipitation significantly restrained the soil respiration, and the effect weakened as
the soil water faded away (Fig. 9), which was the same pattern as with the effect of
SWC on soil respiration in the wetting-drying cycle affected by irrigation. This
finding means that irrigation and precipitation both affect the soil respiration by
affecting the SWC, which affects the activities of the root and microorganisms and the
diffusion of $O_2$ and the solute (Luo & Zhou, 2006). The soil temperature was also
affected by the change in soil moisture. To affect soil respiration, for example, the
precipitation took one day for the soil respiration to recover from the restrain to a
normal level, while irrigation took four days to recover (Fig. 6, Fig. 8). This
difference occurred because the drip irrigation decreased the soil temperature much
more than the precipitation did as the irrigation water was taken directly from a deep
well which was colder than the precipitation water. Therefore, the effect of soil water
on soil respiration was always confounded by the soil temperature (Davidson *et al.*,
641  1998).
Our results show that the 12 mm precipitation had little effect on the soil moisture
and soil respiration. The 37.8 mm precipitation resulted in soil respiration in the mf
and nmf fields because the precipitation can directly infiltrate into soil in the furrows.
However, this precipitation event restrained soil respiration in the mr and nmr because
the precipitation cannot infiltrate into the soil in the mr but can infiltrate into the nmr.
This difference led the soil moisture in the mr still to decrease without irrigation and
the soil moisture in the nmr to be very high and restrain soil respiration. After the 48
mm precipitation, the soil respirations were all restrained in the ridges and furrows in
the mulched and non-mulched fields as the SWCs were all approaching 0.3 (Fig. 8).
The above arguments indicate that the effect of precipitation on the soil respiration
was determined by the SWC. As the SWC is related to the precipitation amount, the
amount and timing of the precipitation affected the soil respiration by affecting the
SWC.
The hydrological responses of precipitation in the field were changed by the





plastic mulch and its physical non-permeability to water. Moreover, this barrier was
the reason the precipitation effect on the soil respirations was different in the mulched
and non-mulched fields. For example, the soil respiration in the nmr was larger than
in the mf and mh during 2016. However, the result was contrary in 2014 and 2015.
With little rainfall during 2014 and 2015, the soil moisture in the mf was larger than
in the nmr (Fig. 5). Additionally, the strong evaporation in the nmr without the plastic
mulch protection and the fact that the soil moisture in the mr can horizontally
infiltrate into the mf are considered. The soil temperature in the mf was also larger
than in the nmr (Fig. 7c). These two factors determined that the soil respiration in the
nmr was smaller than in the mf and mh. With more rainfall during 2016, the soil
moisture in the nmr was larger than in the mf considering that the rainfall cannot
penetrate the plastic mulch. Moreover, their temperatures were not as different as with
the effect of irrigation, so the soil respiration in the nmr was larger than in the mf and
mh during 2016. The precipitation resulted in greater soil respiration in the
non-mulched field than in the mulched field, and the amount of soil respiration from
2014 to 2016 increased. Therefore, we can speculate the magnitude at which the
mulch accelerating soil respiration was related to the precipitation amount.
673        Although the precipitation restrained the soil respiration at a high SWC right after
precipitation, the restrain was quickly depleted. Therefore, the precipitation increased
the soil respiration in the mulched and non-mulched fields by improving soil moisture
conditions during the growing season, particularly in an arid area. Moreover, on a
global scale, the soil respiration rates were found to be positively correlated with the
mean annual precipitation (Raich & Schlesinger, 1992) and the soil respiration
increased linearly with the mean annual precipitation (Zhou *et al.*, 2009).

## 680    5. Summary and Conclusions

681        Plastic mulch is now widely used in agriculture around the world due to the
continuous fall in the prices of plastic products and increasing development of plastic
industries, particularly in developing countries, such as China. The changing land
cover with a mass of the PFM field will affect the energy, water and carbon cycle
regionally or globally. However, how plastic mulch affects $CO_2$ emissions in an
agriculture field remains unclear. This uncertainty is particularly pronounced in arid
areas under the condition of climate changes, such as rising temperatures and shifting
precipitation, which both have severe effects on the soil carbon balance.
689        A comparative experiment was conducted in a plastic mulch drip irrigation field in
an arid area of Northwest China to detect how the soil respiration is affected by
plastic mulch, irrigation and precipitation. The spatial heterogeneity of the
microclimate and soil respiration was enhanced by the plastic mulch. Crop growth
was improved with the improved environmental conditions of the soil temperature
and moisture, which increase respiration of roots and microorganisms with a greater
mineralization and higher litter fall and root biomass. The furrows, plant holes and
plastic mulch were three important pathways for $CO_2$ emissions in the mulched field.




The relationship between the soil respiration and soil temperature was weakened by
frequent irrigation and precipitation. The soil respiration was first restrained and then,
enhanced in a wetting-drying cycle caused by irrigation and precipitation. The soil
respiration in the mulched field was larger than in the non-mulched field, both in the
ridges and furrows during the growing season. This result indicated that the plastic
mulch increased the soil respiration in an arid area. However, it was observed that the
magnitude of the plastic mulch accelerating soil respiration decreased with the
amount of precipitation over three years. Both irrigation and precipitation controlled
the seasonal variation in soil respiration in the mulched field in the arid area. However,
irrigation had the same effect on the soil respiration in the mulched and non-mulched
fields as the drip tapes that were beneath the plastic mulch, while precipitation
primarily affects the soil respiration in the non-mulched field because of the mulch
barrier to precipitation. Moreover, a linear relationship was found between the
differences in the soil respiration of the mulched and non-mulched fields and the
precipitation amount by collecting other studies. With increased precipitation, the
function of the plastic mulch accelerating soil respiration was weakened. This
outcome indicates whether the plastic mulch increasing soil respiration depends on
the climate. In an arid area, the plastic mulch will increase the soil respiration. In a
humid area, the mulch will decrease the soil respiration compared to the non-mulched
field because precipitation increases the soil respiration more in the non-mulched field
than in the mulched field.
On the one hand, the plastic mulch will improve crop growth. However, the
approach will also increase $CO_2$ emissions in an arid area with the increase being
altered by precipitation in the field. With extreme precipitation and the rapid
expansion of the PFM field from natural ecosystems recently occurring in the
Xinjiang Uygur Autonomous Region, the challenges for controlling greenhouse gas
emissions in the arid area is still severe. Plastic mulch and irrigation should be better
depicted in future soil carbon models. Linking the hydrologic and Carbon cycles via
the conservation of water resources is crucial for improving agronomic yields and soil
C sequestration in dryland (Lal, 2004).

## Acknowledgement

This research was support by the Ministry of Science and Technology (2016YFC0402701,
2016YFA0601603), the National Science Foundation of China (NSFC 91647205) and the
Foundation of the State Key Lab- oratory of Hydroscience and Engineering of Tsinghua University
(2016-KY-03). We gratefully appreciate their support. We acknowledge the staff at Tsinghua
Universtiy-Korla Oasis Eco-hydrology Experimental Research Station for their kindly help and
assistant. Also, the authors thank Dr. Mohd Yawar Ali Khan for the help with language
improvement.





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

Methods and Fertilization Measures  on Soil Respiration and Its Component Contrib.
Scientia Agricultura Sinica, **45**, 2420-2430.
Raich JW, Schlesinger WH (1992) The global carbon dioxide flux in soil respiration and its relationship
to vegetation and climate. Tellus B, **44**, 81-99.
Raich JW, Tufekciogul A (2000) Vegetation and soil respiration: Correlations and controls.
Biogeochemistry, **48**, 71-90.
Reichstein M, Beer C (2008) Soil respiration across scales: The importance of a model–data integration
framework for data interpretation. Journal of Plant Nutrition and Soil Science, **171**, 344-354.
Tarara JM (2000) Mircroclimate modification with plastic mulch. HortScience, **35**, 180.
Tian F, Yang P, Hu H, Dai C (2016) Partitioning of Cotton Field Evapotranspiration under Mulched Drip



821    Irrigation Based on a Dual Crop Coefficient Model. Water, **8**, 72.

822 Wang YP, Li XG, Fu T, Wang L, Turner NC, Siddique KHM, Li F-M (2016) Multi-site assessment of the

823    effects of plastic-film mulch on the soil organic carbon balance in semiarid areas of China.

824    Agricultural and Forest Meteorology, **228–229**, 42-51.

825 Xiang G, Gong D, Fengxue G (2014) Inhibiting soil respiration and improving yield of spring maize in

826    fields with plastic film mulching. Transaction of the Chinese Society of Agricultural

827    Engineering, **30**, 62-70.

828 Xu L, Baldocchi DD, Tang J (2004) How soil moisture, rain pulses, and growth alter the response of

829    ecosystem respiration to temperature. Global Biogeochemical Cycles, **18**, n/a-n/a.

830 Yaghi T, Arslan A, Naoum F (2013) Cucumber (Cucumis sativus, L.) water use efficiency (WUE) under

831    plastic mulch and drip irrigation. Agricultural Water Management, **128**, 149-157.

832 Yan M, Zhou G, Zhang X (2014) Effects of irrigation on the soil $CO_2$ efflux from different poplar clone

833    plantations in arid northwest China. Plant and Soil, **375**, 89-97.

834 Yang P, Hu H, Tian F, Zhang Z, Dai C (2016) Crop coefficient for cotton under plastic mulch and drip

835    irrigation based on eddy covariance observation in an arid area of northwestern China.

836    Agricultural Water Management, **171**, 21-30.

837 Yu Y, Zhao C, Stahr K, Zhao X, Jia H, De Varennes A (2016) Plastic mulching increased soil

838    $CO_2$concentration and emissions from an oasis cotton field in Central Asia. Soil Use and

839    Management, **32**, 230-239.

840 Zeng N, Zhao F, Collatz GJ, Kalnay E, Salawitch RJ, West TO, Guanter L (2014) Agricultural Green

841    Revolution as a driver of increasing atmospheric $CO_2$ seasonal amplitude. Nature, **515**,

842    394-397.

843 Zhang Z, Hu H, Tian F, Yao X, Sivapalan M (2014) Groundwater dynamics under water-saving irrigation

844    and implications for sustainable water management in an oasis: Tarim River basin of western

845    China. Hydrology and Earth System Sciences, **18**, 3951-3967.

846 Zhang Z, Tian F, Hu H (2011) Spatial and temporal pattern of soil temperature in cotton field under

847    mulched drip irrigation condition in Xinjiang. Trans. CSAE, **2011**, 44-51.

848 Zhou X, Talley M, Luo Y (2009) Biomass, Litter, and Soil Respiration Along a Precipitation Gradient in

849    Southern Great Plains, USA. Ecosystems, **12**, 1369-1380.

850