# Peer review of "Precipitation alters plastic film mulching impacts on soil respiration in an arid area of northwest China Authors: Guanghui Ming1, Hongchang Hu1, Fuqiang Tian1\*, Zhenyang Peng1, Pengju Yang1, Yiqi Luo2, 3 Affiliations: 1De"

_Hydrology and Earth System Sciences, 2017_

## Referee Comment (RC1) · Anonymous Referee #1 · 31 Aug 2017

General Comments

The paper titled "Precipitation alters plastic film mulching impacts on soil respiration in an arid area of Northwest China" investigates the influence of plastic film mulch on soil respiration and illuminates the complex relationship precipitation has on these processes. The paper is generally well written, of sufficient scientific quality and of interest for publication in Hydrology and Earth System Sciences, with some modification (see below). However, there are several larger issues with the structure of the paper that need to be addressed. Chiefly, the discussion does not integrate the results of the study and discuss the implications of the work on the broader literature and "real world"

[Figure]

scenarios in enough detail. One critical question that I do not believe has been sufficiently answered is "what is the importance of the work?". This is a critical question that needs to be clearly addressed in the discussion. Additionally, there are also several issues with flow and brevity in the manuscript, particularly the introduction and discussion, that needs to be addressed. I suggest a re-writing/structuring of the discussion to improve the flow of the manuscript and further discussion on the implications of their results to the broader community.

Specific Comments

Introduction: The introduction is quite long and should be condensed, only keeping what is critical to the paper. Ideally, the introduction would be shortened by ~1 page. To that end, I suggest condensing the temperature and moisture effects into 1 paragraph. Additionally, the first paragraph is very long, I suggest separating them into two paragraphs at line 67, as it is a natural break in the topics from general background to China focused.

Results: Throughout the results there are statements that belong in the discussion, particularly where the authors state a reason for a given trend. For instance, P22L496-513 are relating the current work to past work, while giving the regional/climatic context. This paragraph is not results but discussion. Please move these sentences/paragraphs, and others like it, to the discussion. Additionally, there are many instances of the results being repeated in the discussion (see comment below).

The analysis between other studies of differing climates (P22L496-513) is interesting but needs further clarification and integration into the discussion. Furthermore, it would be very instructive to have each of the studies highlighted in Figure 10, where each point is displayed as a unique symbol or colour to help the reader differentiate which point is from which study.

Discussion: There are several sentences in the discussion where the results are being restated (e.g., "Our results indicate that the soil respiration in the ridges of the mulched

field (mr) as measured by uncovering the plastic mulch was greater than in the furrows (mf). This finding indicates that much of the $CO_2$ gathers beneath the plastic mulch because of the plastic barrier.") Although at times this is useful, I suggest removing these sentences to help with flow and brevity of the manuscript, which is quite long. Alternatively, these sentences can be reworded to combine both the results and the implications, which would greatly improve the flow of the discussion. For instance, the above quotation could be rewritten as "Soil respiration in the ridges of the mulched field (mr) was much greater than in the furrows (mf), indicating that $CO_2$ gathers beneath the plastic mulch barrier". Please go through the manuscript and adjust instances of this issue. The discussion needs to be significantly shortened; however, this can be mostly achieved through removing repetition of results.

How do your fitted respiration/soil temperature or SWC curves compare to the literature? Are your values representative of the broader literature? This would be a useful comparison in the discussion (particularly section 4.2).

Although the discussion is long, it lacks a strong comparison of the results to global trends, nor does it integrate the components (temperature, soil moisture, etc.) of soil respiration adequately. There needs to be a final paragraph that integrates the effects of plastic much on soil respiration and the broader implications of the work. Within this paragraph it is critical to highlight what the new findings of this study are within the context of the broader literature.

Conclusions: In the conclusion, the authors state that there was increased mineralization, yet this was not discussed in the discussion. It would be prudent to include a short discussion of the effects of the plastic mulch in this context, perhaps including it in the above discussion.

Technical Comments

P1L33 missing comma before "while" P3L113-117 This sentence is awkward and long, I suggest breaking the sentence into two separate sentences. P9L295 remove the

"-" after 10 P21L477 remove the "s" from Soil respirations P21L448 missing comma before "although" P22L494 please provide the statistics for the use of significantly. If statistics were not used to quantify the significance of the relationship, please remove significantly as it specifically refers to a statistical significance. P23L526 missing space between "the" and "accelerated" P24L533-534 this is an awkward sentence, I suggest rewording to "This finding indicates that CO2 from soil respiration gathers beneath the plastic mulch barrier, resulting in a greater flux of CO2." P24L563 the period at the end of the sentence is misplaced. P25L574-581 these sentences are a repeat of results, please delete. P25L574-594 this paragraph does not add to the manuscript at simply restates the results and part of the introduction. I suggest removing the paragraph. P25L596 please remove the "-" between "high water content" P26L608-609 awkward sentence, please reword to "There was poor correlation between soil respiration and the soil temperature in different parts of the mulched and non-mulched fields". P26L627-630 this is a run-on sentence, please revise into two separate sentences P27L657 please remove the "s" from respirations

Figure 3: it is unclear from your methods why certain datasets are left out of these graphs (e.g., mp in 2014 and 2015). Was the data not collected? Please clarify this discrepancy between your methods, which states that each was measured for all years, and your display of the data in the text of the methods or results.

---

## Referee Comment (RC2) · Anonymous Referee #2 · 1 Dec 2017

This manuscript reports findings from a field experiment on the effect of plastic mulching on $CO_2$ emissions from soil furrows and ridges, in relationship to temperature and soil moisture. This topic is of interest to readers of HESS. However the presentation of the results and discussion in this manuscript is unclear, making it difficult to interpret and evaluate the findings. Furthermore, the final conclusions are not supported by a critical evaluation of the uncertainty and statistical power of the results.

Specific comments: - The introduction is very long and contains a lot of unnecessary information

- Consistent references to figures are missing from the text
- Throughout the paper, reference is made to the seasonal respiration. However, it is not clearly defined what is meant with this. I assume it refers to the growing season. However, how many days was this exactly? Was the length of the season the same number of days each year? How was this decided?

- Fig 3: why was $CO_2$ respiration measured for a different number of days in the different years? This should be addressed in the methods and results section.

- Fig 7: Where do the data of soil respiration with days after irrigation come from? The method section states that respiration was measured every 2 weeks.

- Page 22, line 500-518. Looking at the figure, the data points form a cloud with one outlier. It is not appropriate to assume a linear correlation here.

- The sections on the effect of irrigation and precipitation on soil respiration can be combined as both seem to produce similar effects

- Though the English grammar is good, the argumentation and writing throughout the manuscript is hard to follow and needs careful editing.

- A discussion of the statistical significance and uncertainty in the findings reported here is missing. such an evaluation would be essential here to support their broader claim that plastic mulch increased $CO_2$ emissions in arid environments.

In addition, all figures, tables and their headings need a lot of improvement:

- Fig. 1 is very hard to read in color and unreadable when printed in black and white. A schematic figure may be clearer and more helpful

- Throughout the manuscript, figures and table headings are missing definitions of abbreviations and labels of the different treatments

- Figure 3 is very unclear The labeling on the y-axis of the top and bottom figure is missing. The scales of the x-axis and y-axis of the 3 graphs are different, making it impossible to compare the data. Furthermore, the layout is inconsistent between

graphs.

- Figure 4: the layout of this figure is confusing, and hard to read when printed in black and white. In addition, the figure heading states this is the seasonal accumulative soil respiration, whereas the figure shows years. Define this more clearly. Which season is considered here?

- Fig 7: The temperature plot is not needed here, removing it may make the other 2 figures more readable.

- Figure 9: what is G? Why are dates reported here, when other figures use days?

- If I understood the text correctly (page 22, line 500-518), this figure was made using literature values. This should be explicitly stated in the figure heading, and references should be included. Also, define dF in figure heading.

---

## Author Comment (AC1) · 6 Dec 2017

**Guanghui Ming**

Authors' responses (blue color) start with "response".

The paper titled "Precipitation alters plastic film mulching impacts on soil respiration in an arid area of Northwest China" investigates the influence of plastic film mulch on soil respiration and illuminates the complex relationship precipitation has on these processes. The paper is generally well written, of sufficient scientific quality and of interest for publication in Hydrology and Earth System Sciences, with some modification (see below). However, there are several larger issues with the structure of the paper that need to be addressed. Chiefly, the discussion does not integrate the results of the study and discuss the implications of the work on the broader literature and "real world" scenarios in enough detail. One critical question that I do not believe has been sufficiently answered is "what is the importance of the work?". This is a critical question that needs to be clearly addressed in the discussion. Additionally, there are also several issues with flow and brevity in the manuscript, particularly the introduction and discussion, that needs to be addressed. I suggest a re-writing/structuring of the discussion to improve the flow of the manuscript and further discussion on the implications of their results to the broader community

Response: Thanks the anonymous referee a lot for the constructive comments that helped us to improve the quality of our manuscript. We have restructured our manuscript and rewritten some sentences to broaden the research meaning and make it more logical and readable as the referee suggested. Below we address all the comments on a point-by-point basis.

Specific Comments:

Introduction:

The introduction is quite long and should be condensed, only keeping what is critical to the paper. Ideally, the introduction would be shortened by _1 page.

Response: Done. The introduction has been shortened by about 1 page with moving "the effect of different climates on soil respiration" paragraph to the discussion, and removing other non-relevant sentences.

To that end, I suggest condensing the temperature and moisture effects into 1 paragraph.

Response: Done. The temperature and moisture effect paragraphs have been condensed into 1 paragraph.

Additionally, the first paragraph is very long, I suggest separating them into two paragraphs at line 67, as it is a natural break in the topics from general background to China focused.

Response: Done. The first paragraph has been separated into two paragraphs as per the referee's suggestion. Also, the sentence about the expansion of PFM cultivation around the world has been moved to the second paragraph to ensure that the first paragraph focuses on anthropogenic effect and the second paragraph focuses on the specific PFM effect.

Results:

Throughout the results there are statements that belong in the discussion, particularly where the authors state a reason for a given trend. For instance, P22L496-513 are relating the current work to past work, while giving the regional/climatic context. This paragraph is not results but discussion. Please move these sentences/paragraphs, and others like it, to the discussion. Additionally, there are many instances of the results being repeated in the discussion (see comment below).

Response: The paragraph has been moved into the discussion. The repeated instances of the results have been removed.

The analysis between other studies of differing climates (P22L496-513) is interesting but needs further clarification and integration into the discussion. Furthermore, it would be very instructive to have each of the studies highlighted in Figure 10, where each point is displayed as a unique symbol or color to help the reader differentiate which point is from which study.

Response: Done. The paragraph has been clarified and moved into the discussion as per the reviewer's suggestion. The climate and reference of each study has been highlighted in Figure 10.

Discussion:

There are several sentences in the discussion where the results are being restated (e.g., "Our results indicate that the soil respiration in the ridges of the mulched field (mr) as measured by uncovering the plastic mulch was greater than in the furrows (mf). This finding indicates that much of the $CO_2$ gathers beneath the plastic mulch because of the plastic barrier.") Although at times this is useful, I suggest removing these sentences to help with flow and brevity of the manuscript, which is quite long. Alternatively, these sentences can be reworded to combine both the results and the implications, which would greatly improve the flow of the discussion. For instance, the above quotation could be rewritten as "Soil respiration in the ridges of the mulched field (mr) was much greater than in the furrows (mf), indicating that $CO_2$ gathers beneath the plastic mulch barrier". Please go through the manuscript and adjust instances of this issue. The discussion needs to be significantly shortened; however, this can be mostly achieved through removing repetition of results.

Response: Thanks for your patient guidance and the constructive advice. We have

reworded the discussion with removing repetition of results and combining the results and the implications throughout the manuscript.

How do your fitted respiration/soil temperature or SWC curves compare to the literature? Are your values representative of the broader literature? This would be a useful comparison in the discussion (particularly section 4.2).
Response: We have compared our results about the soil temperature and moisture sensitivity of soil respiration with other studies around the world.

Although the discussion is long, it lacks a strong comparison of the results to global trends, nor does it integrate the components (temperature, soil moisture, etc.) of soil respiration adequately. There needs to be a final paragraph that integrates the effects of plastic mulch on soil respiration and the broader implications of the work. Within this paragraph it is critical to highlight what the new findings of this study are within the context of the broader literature.
Response: We have added another paragraph by combing our new findings and the implications to the global trends.

Conclusions:
In the conclusion, the authors state that there was increased mineralization, yet this was not discussed in the discussion. It would be prudent to include a short discussion of the effects of the plastic mulch in this context, perhaps including it in the above discussion.
Response: The effect of plastic mulch on mineralization has been added into the first paragraph of discussion.

Technical comments:
P1L33 missing comma before "while".
Response: Done. The missing comma has been added.

P3L113-117 this sentence is awkward and long, I suggest breaking the sentence into two separate sentences.
Response: Done. The sentence has been reworded into two separate sentences as per the referee's suggestion.

P9L295 remove the "-"after 10
Response: Done.

P21L477 remove the "s" from Soil respirations.
Response: Done.

P21L478 missing comma before "although"
Response: Done.

P22L494 please provide the statistics for the use of significantly. If statistics were not

used to quantify the significance of the relationship, please remove significantly as it specifically refers to a statistical significance.
Response: Done.

P23L526 missing space between "the" and "accelerated"
Response: Done.

P24L533-534 this is an awkward sentence, I suggest rewording to "This finding indicates that $CO_2$ from soil respiration gathers beneath the plastic mulch barrier, resulting in a greater flux of $CO_2$."
Response: Done. This sentence has been reworded as per the referee's suggestion.

P24L563 the period at the end of the sentence is misplaced.
Response: Done. The period has been properly placed.

P25L574-581 these sentences are a repeat of results, please delete.
Response: Done.

P25L574-594 this paragraph does not add to the manuscript at simply restates the results and part of the introduction. I suggest removing the paragraph.
Response: Done.

P25L596 please remove the "-"between "high water content".
Response: Done.

P26L608-609 awkward sentence, please reword to "There was poor correlation between soil respiration and the soil temperature in different parts of the mulched and non-mulched fields".
Response: Done.

P26L627-630 this is a run-on sentence, please revise into two separate sentences.
Response: Done.

P27L657 please remove the "s" from respirations.
Response: Done.

Figure 3: it is unclear from your methods why certain datasets are left out of these graphs (e.g., mp in 2014 and 2015). Was the data not collected? Please clarify this discrepancy between your methods, which states that each was measured for all years, and your display of the data in the text of the methods or results.
Response: The missed datasets in 2014 and 2015 were due to no observation. This has been clarified in methods and the Figure heading.

---

## Author Comment (AC2) · 6 Dec 2017

**Guanghui Ming**

Authors' responses (blue color) start with "response".

-This manuscript reports findings from a field experiment on the effect of plastic mulching on $CO_2$ emissions from soil furrows and ridges, in relationship to temperature and soil moisture. This topic is of interest to readers of HESS. However, the presentation of the results and discussion in this manuscript is unclear, making it difficult to interpret and evaluate the findings. Furthermore, the final conclusions are not supported by a critical evaluation of the uncertainty and statistical power of the results.

Response: Thanks a lot for the reviewer's constructive comments that helped us to improve the quality of our manuscript. We have restructured/rewritten our results and discussion as per the reviewer's suggestions and the uncertainty was evaluated with statistical analysis to support our results. Below we address all the comments on a point-by-point basis.

Specific comments:
- The introduction is very long and contains a lot of unnecessary information

Response: The introduction has been shortened by about 1 page with moving "the effect of different climates on soil respiration" paragraph to the discussion, and removing other non-relevant sentences. Thanks.

- Consistent references to figures are missing from the text

Response: The missing references have been added in the text according to the reviewer's suggestions. Thanks.

- Throughout the paper, reference is made to the seasonal respiration. However, it is not clearly defined what is meant with this. I assume it refers to the growing season. However, how many days was this exactly? Was the length of the season the same number of days each year? How was this decided?

Response: The seasonal respiration refers soil respiration in the cotton growing season. The planting date of the cotton depends on the local temperature and thus differs in the three experimental years. Generally, the growing season is from germination in late April to harvest in late September, about 150 days. We have addressed it in the revised manuscript.

- Fig 3: Why was $CO_2$ respiration measured for a different number of days in the different years? This should be addressed in the methods and results section.

Response: Study periods were concentrated on the growing season as soil respiration in non-growing seasons is extremely low. The experiments in the year 2014 and 2015 began in the bud stage when cotton began to grow faster. Therefore, the lengths of measured periods are different for the three years, with 95, 60, 100 days, respectively. We have addressed it in the methods and results section. Thanks.

- Fig 7: Where do the data of soil respiration with days after irrigation come from? The method section states that respiration was measured every 2 weeks.

Response: We are sorry for the misleading. Actually, soil respiration experiments were carried out randomly between two irrigation events, i.e., the measurement day can be any day after an irrigation event. We revised the corresponding sentences accordingly. Thanks.

- Page 22, line 500-518. Looking at the figure, the data points form a cloud with one outlier. It is not appropriate to assume a linear correlation here.

Response: Inspired by the comment, we have reviewed more literatures and found one more experimental result in subtropical monsoon climate with precipitation larger than 1000 mm, which is shown in the figure below. We also carried out the statistical test to support the linear correlation conclusion. Thanks.

[Figure]

Figure. The relationship of the difference of soil respiration between mulched and non-mulched fields with precipitation in different climates. (dF means the soil respiration in non-mulched field minus that in mulched field; In the five points of arid areas, two points from (Yu *et al*., 2016) are in the circle while the other three points from our research are outside of the circle.)

- The sections on the effect of irrigation and precipitation on soil respiration can be combined as both seem to produce similar effects

Response: Irrigation and precipitation both affected soil respiration by affecting soil moisture. But drip irrigation and precipitation had different effects on soil moisture in mulched and non-mulched fields. So we separately analyzed and discussed the effect of irrigation and precipitation on soil respiration in order to get our main findings. We would prefer to keep the two sections as they are. Thanks.

- Though the English grammar is good, the argumentation and writing throughout the manuscript is hard to follow and needs careful editing.
Response: Several paragraphs in the manuscript have been carefully restructured and some of the argumentation has been formatted to make it more logical and readable. Thanks.

- A discussion of the statistical significance and uncertainty in the findings reported here is missing. such an evaluation would be essential here to support their broader claim that plastic mulch increased $CO_2$ emissions in arid environments.
Response: Thanks for your valuable suggestion. The statistical significance of soil respiration in mulched and non-mulched field has been assessed with the hypothesis $t$-test. The uncertainty has been assessed with the standard derivation of duplicates. The results show that our conclusion can be supported by the evaluation that plastic mulch increased $CO_2$ emissions in arid environments.

In addition, all figures, tables and their headings need a lot of improvement:

- Fig. 1 is very hard to read in color and unreadable when printed in black and white. A schematic figure may be clearer and more helpful
Response: We have re-drawn a new schematic figure to illustrate cotton planting and drip pipe arrangement, as well as the experiment design.

- Throughout the manuscript, figures and table headings are missing definitions of abbreviations and labels of the different treatments
Response: Revised. Thanks.

- Figure 3 is very unclear. The labeling on the y-axis of the top and bottom figure is missing. The scales of the x-axis and y-axis of the 3 graphs are different, making it impossible to compare the data. Furthermore, the layout is inconsistent between graphs.
Response: The labeling, scales and layout of the three graphs has been rearranged to make them more readable. Thanks.

- Figure 4: the layout of this figure is confusing, and hard to read when printed in black and white. In addition, the figure heading states this is the seasonal accumulative soil respiration, whereas the figure shows years. Define this more clearly. Which season is considered here?
Response: The data shown in the figure are the seasonal accumulative soil respiration.

The labels of x-axis show different years indicating the growing seasons of these years. The growing season in our paper can be generally considered from late April to late September, as described in the above response to Specific Comment #3. The authors are sorry for the confusion, and the manuscript has been revised accordingly. Thanks.

- Fig 7: The temperature plot is not needed here, removing it may make the other 2 figures more readable.
Response: Done. Thanks.

- Figure 9: what is G? Why are dates reported here, when other figures use days?
Response: We are sorry for the carelessness. Indeed, "G" is a drawing mistake. It has been deleted in the revised manuscript. The dates have been replaced by DOY (days of the year) according to the suggestion. Thanks.

- If I understood the text correctly (page 22, line 500-518), this figure was made using literature values. This should be explicitly stated in the figure heading, and references should be included. Also, define dF in figure heading.
Response: Yes, figure 10 is the analysis of the literature review. As per the suggestion of the reviewer's, the figure heading has been explicitly stated with detailed dF definition. References have also been added for each study. Thanks.

---

## Author Response (AR2)

The file contains three parts: Response to anonymous references, marked-up version to the referee's comments and marked-up version polished by a qualified company.

Authors' responses (blue color) start with "response".
Response to Anonymous Reference#1

The revision of the manuscript titled "Precipitation alters plastic film mulching impacts on soil respiration in an arid area of Northwest China" focused on the effect of plastic film mulching on CO2 respiration and put their findings into the global climate context. The science is of sufficient quality for publication, while some grammatical issues remain (see minor comments below). The discussion is generally well done and well cited, however, there are two areas where the authors could improve their manuscript through relatively minor revisions/additions (see major comments below). Both of these revisions/additions are to improve the context of their findings to the broader community/literature/issues. Once the major and minor comments have been addressed, along with a thorough grammatical edit, I believe the manuscript will suitable for publication.

Response: We thank the reviewer for these detailed and relevant comments that will improve the overall quality of this manuscript. A qualified language company was hired to polish the manuscript all over again. We have outlined our response to each of the comments below.

Major Comments

P388 – 399 What are the broader implications of using the different film/mulch features? Please expand on this paragraph to discuss the importance of plastic selection.

Response: Thanks for your suggestion. We have added a short paragraph in Line 427-433 to discuss the effect of plastic film on both soil $CO_2$ emission and soil fertility. High-density plastic film is recommended for reducing soil $CO_2$ emissions and plastic film residues despite its higher price.

L468 This is an interesting point and an interesting analysis below that does put the results, and conflicting results of other studies, into context. However, I believe that the discussion does not put it in the context of the larger issues; chiefly, depending on annual/seasonal climate (precipitation) is PMF recommended from a CO2 emissions perspective? I strongly suggest the authors add a short discussion on this issue to link their work with the broader community/issue of PFM on a global context. You have 1 sentence (L501-502) in the conclusions that should be expanded in the discussion as it is a very interesting finding (even if it is preliminary).

Response: As per your suggestion, a more detailed discussion has been added to the manuscript in Line 537-544 to expand our results to the global context. Below is the added discussion paragraph:

*Based on the relationships between precipitation and soil respiration in the PFM fields obtained above, plastic film mulching is recommended for application in areas with precipitation greater than 200 mm, i.e., semi-arid and humid areas, to decrease soil $CO_2$ emissions and increase soil carbon sequestration. Decreasing soil $CO_2$ emissions indicates increasing soil organic carbon and maintaining soil fertility to obtain a stable yield. Our results are consistent with those of Zhang et al. (2018), who concluded that PFM where precipitation is greater than 230 mm can result in a stable crop yield on the Loess Plateau.*

Minor Comments

L23 replace "this" with "these"

Response: Correction was made as per your suggestion

L23 replace "the" with "a"

Response: Correction was made as per your suggestion

L30 What is the $CO_2$ flux from the furrow and planting hole? Please provide this for comparison to the plastic mulch

Response: Correction was made as per your suggestion

L57 missing "the" before "intensive"

Response: Correction was made as per your suggestion

L59 remove comma and replace "which" with "that"

Response: Correction was made as per your suggestion

L62 What is the global PFM usage as a percent of arable land? Please include this value so the study can be placed better in the global context

Response: Yes, we agree with this suggestion. The value has been added.

L73-74 replace ", which, however," with "that"

Response: Correction was made as per your suggestion

L77-78 of which study? Please clarify

Response: We thank the reviewer for this suggestion, the sentenced has been rewritten in Line 81-82

L101 It is improper grammar to start a sentence with an abbreviation. Please change any occurrences of this throughout the manuscript.

Response: As per your suggestion, correction was made in the whole manuscript.

L128 Please provide the full name of the evaporation pan used and not the abbreviated symbol.

Response: Correction was made as per your suggestion

L125 please italicise species names

Response: Correction was made as per your suggestion

L152 missing comma before "as"

Response: Correction was made as per your suggestion

L170 replace ", i.e.," with a colon

Response: Correction was made as per your suggestion

L252 missing comma before "although"

Response: Correction was made as per your suggestion

L363 change "motivated" to "increased"

Response: Correction was made as per your suggestion

L361-365 Please provide the DOY beside each event because it is unclear which event you are referring to on the graph. Currently, it is unclear which event is associated with which trend and is difficult to properly evaluate this paragraph.

Response: Thanks for your suggestion, DOY beside each event has been added.

L383 Please provide values for CO2 emitted.

Response: Correction was made as per your suggestion

L432-434 The quadratic (parabolic) SWC Rs relationship is well documented throughout the literature. I suggest adding a few references and maybe a sentence detailing this wide-spread observation.

Response: As per your suggestion, references and sentences have been added.

L438-446 Yes, this relationship has been described as these other equations but as you point out it is typically due to another limiting factor and do not represent the full SWC Rs function. I would add a statement after line 444 to this effect to ensure that the reader is aware of this issue.

Response: Thanks for your suggestion, the statement has been added.

Response to Anonymous Reference#2

The manuscript has been much improved from its previous version. The data representation and figures are now better and the focus and message of the manuscript is clear. The findings of the authors demonstrate the interplay between precipitation, irrigation and their effect on soil respiration under plastic mulching. Though it is perhaps not surprising that these effects are spatially heterogeneous and depend on soil moisture, these aspects of this management strategy have not much been highlighted before. Therefore, I think this manuscript will be a valuable contribution to HESS.

Response: We thank the reviewer for the detailed and relevant comments that will improve the overall quality of the manuscript. Below we address all the comments on a point-by-point basis.

The manuscript could be improved by providing more detailed information on how the average and cumulative respiration rates were calculated. I.e. report the area ratios for the different parts of the field, and what was the daily variability in respiration measured?

Response: Thanks for your suggestion. We have added in Line 206-217 for details of calculating methods for average and cumulative respiration rate and the area ratio for various parts of the field. The daily variability in respiration measured is represented by error bar of daily average respiration in Fig. 3.

Line 383-395: In this part of the discussion, the authors seem to make conclusions about the CO2 emission from different types of plastic mulch. Though this is an interesting part of the manuscript, it is not clear to me how these results follow from the data reported in this manuscript. Is this based on literature information? Or was this part of the findings of this study? This section should be revised to make this more clear.

Response: Indeed, this part is based on literature information to provide reference for plastic film selection in agriculture, considering our research only has one plastic film type. We discussed the effect of plastic film both on soil $CO_2$ emission and soil fertility. High-density plastic film is recommended for reducing soil CO2 emissions and plastic film residues despite its higher price.

The writing of the manuscript has been much improved, but there are a few sections that could benefit from a careful language check. For example, but not limited to:

line 37: remove 'the'

Response: Correction has been made as per your suggestion.

line 83-84: check sentence, replace 'import' with 'important'

Response: Correction has been made as per your suggestion.

Line 96: remove 's' from attentions

Response: Correction has been made as per your suggestion.

line 180: check sentence

Response: The sentence has been carefully checked and revised.

Line 366-373: this section is difficult to follow. Check and consider revising

Response: This section has been carefully checked and revised.

line 437 add: 'findings by' in front of 'Wang et al….'

Response: Correction has been made as per your suggestion.

[revised manuscript text omitted]
 plant holes, the plastic mulch was inserted into the soil covering two plant holes, and Sscotch Ttape was used to seal the interspaces between the plastic mulch and collar to prevent air leakage. To measure the $CO_2$ emissions through the plastic mulch, PVC collars were buried into the soil under the mulch, with Sscotch tTape sealing the interspaces. Detailed measurement methods are further described in Berger et al. (2013)

.

The soil temperature and soil moisture adjacent to each PVC collar at a depth of 5 cm were monitored  using the auxiliary sensors of the LI-8100A,  concurrently with the soil $CO_2$ flux measurements. The amount of drip irrigation  was  determined  using water meters installed on the branch pipes of the drip irrigation system. The precipitation was measured  using a tipping bucket rain gauge (model TE525MM, Campbell Scientific Inc., Logan, UT, USA), which was mounted 0.7 m above the ground.

**2.3 Data analysis**

The soil respiration from different  areas at a particular time of a day was calculated as the average of three replicates. The daily mean $R_s$ was calculated as the average  $R_s$ measured at various times in a day. The $R_s$ in the mulched ridges was calculated  based on the area ratio of $R_s$ measured through the plant holes and the plastic mulch :

$$R_{r-m} = R_{h-m} * A_{h-m} + R_{p-m} * A_{p-m} \qquad (1)$$

where  $R_{h-m}$ and $R_{p-m}$  represent the soil respiration from the planting hole and plastic mulch, respectively,  and constitute the soil respiration in the ridge ($R_{r-m}$). The term $A$  represents the area ratio of the different parts, and

$A_{h-m}$ and $A_{p-m}$ are 0.3 and 0.7, respectively, in our field.

The seasonal accumulative $R_s$ in the ridges and furrows was calculated by summing the $R_s$ values over the measurement period (Yu et al., 2016;Berger et al., 2013). The

Soil respiration in plastic mulch and non-mulched field was calculated  based on the area ratio of $R_s$ through ridges and furrows :

$$R_m = R_{r-m} * A_{r-m} + R_{f-m} * A_{f-m} \qquad (2)$$

$$R_{nm} = R_{r-nm} * A_{r-m} + R_{f-nm} * A_{f-m} \qquad (3)$$

where $R_m$ and $R_{nm}$  represent soil respiration in mulched and non-mulched field, respectively, and $A_{r-m}$ and $A_{f-m}$ are the area ratio of the ridge and furrow, respectively, which is the same for mulched and non-mulched field 
[revised manuscript text omitted]